# Variation in olfactory neuron repertoires is genetically controlled and environmentally modulated

Ximena Ibarra-Soria[1], Thiago S Nakahara[2], Jingtao Lilue[1], Yue Jiang[3], Casey Trimmer[4], Mateus AA Souza[2], Paulo HM Netto[2], Kentaro Ikegami[3], Nicolle R Murphy[4], Mairi Kusma[1], Andrea Kirton[1], Luis R Saraiva[1], Thomas M Keane[1], Hiroaki Matsunami[3,5], Joel Mainland[4,6], Fabio Papes[2], Darren W Logan[1,4]*

[1]Wellcome Trust Sanger Institute, Cambridge, United Kingdom; [2]Department of Genetics and Evolution, Institute of Biology, University of Campinas, Campinas, Brazil; [3]Department of Molecular Genetics and Microbiology, Duke University Medical Center, Durham, United States; [4]Monell Chemical Senses Center, Philadelphia, United States; [5]Department of Neurobiology, Duke Institute for Brain Sciences, Duke University Medical Center, Durham, United States; [6]Department of Neuroscience, University of Pennsylvania, Philadelphia, United States

**Abstract** The mouse olfactory sensory neuron (OSN) repertoire is composed of 10 million cells and each expresses one olfactory receptor (OR) gene from a pool of over 1000. Thus, the nose is sub-stratified into more than a thousand OSN subtypes. Here, we employ and validate an RNA-sequencing-based method to quantify the abundance of all OSN subtypes in parallel, and investigate the genetic and environmental factors that contribute to neuronal diversity. We find that the OSN subtype distribution is stereotyped in genetically identical mice, but varies extensively between different strains. Further, we identify *cis*-acting genetic variation as the greatest component influencing OSN composition and demonstrate independence from OR function. However, we show that olfactory stimulation with particular odorants results in modulation of dozens of OSN subtypes in a subtle but reproducible, specific and time-dependent manner. Together, these mechanisms generate a highly individualized olfactory sensory system by promoting neuronal diversity.

*For correspondence: dl5@sanger.ac.uk

**Competing interests:** The authors declare that no competing interests exist.

## Introduction

Mapping the neuronal diversity within a brain remains a fundamental challenge of neuroscience. Quantifying variance in a population of neurons within and between individuals first requires precise discrimination of cellular subtypes, followed by an accurate method of counting them. While this has been achieved in a simple invertebrate model containing hundreds of neurons (*White et al., 1986*), applying the same approach to mammalian brains that encompass many millions of neurons represents a significant challenge (*Wichterle et al., 2013*).

The main olfactory epithelium (MOE) is an essential component of the olfactory sensory system. It contains olfactory sensory neurons (OSNs) that express olfactory receptors (ORs), the proteins that bind odorants (*Buck and Axel, 1991*; *Zhao et al., 1998*). The mouse genome codes for over a thousand functional OR genes, but each mature OSN expresses only one abundantly, in a monoallelic fashion (*Hanchate et al., 2015*; *Saraiva et al., 2016*; *Tan et al., 2015*; *Chess et al., 1994*). This results in a highly heterogeneous repertoire of approximately 10 million OSNs (*Kawagishi et al.,*

**eLife digest** Smells are simply chemicals in the air that are recognized by nerves in our nose. Each nerve has a receptor that can identify a limited number of chemicals, and the nerve then relays this information to the brain. Animals have hundreds to thousands of different types of these nerves meaning that they can detect a wide array of smells.

Smell receptors are proteins, and the genes that encode these proteins can be very different in two unrelated people. This could partly explain, for example, why some people find certain odors intense and unpleasant while others do not. However, having different genes for smell receptors does not by itself completely explain why some people are more sensitive than others to particular smells. The amounts of each nerve type in the nose might also differ between people and have an effect, but to date it has not been possible to accurately count them all.

Ibarra-Soria et al. have now devised a new method to essentially count the number of each nerve type in the noses of mice from different breeds. The method makes use of a technique called RNA-sequencing, which can reveal which genes are active at any one time, and thus show how many nerves are producing each type of smell receptor. Ibarra-Soria et al. learned that different breeds of mice had remarkably different compositions of nerves in their noses. Further analysis revealed that this was due to changes to the DNA code near to the genes that encode the smell receptor.

Next, Ibarra-Soria et al. sought to find out how the amount of each nerve type is controlled by giving mice water with different smells for weeks and looking how this affected their noses. These experiments revealed that a small number of the nerve types became more or less common after exposure to a smell. The altered nerves were directly involved in recognizing the smells, proving that the very act of smelling can change the make-up of nerves in a mouse's nose.

These results confirm that the diversity in the nose of each individual is not only dictated by the types of receptors found in there, but also by the number of each nerve type. The next challenge is to understand better how these differences change the way people perceive smells.

*2014*) within the nose of a mouse, stratified into more than a thousand functionally distinct subpopulations, each one characterized by the particular OR it expresses. The monogenic nature of OR expression serves as a molecular barcode for OSN subtype identity. Thus, the MOE offers a unique opportunity to generate a comprehensive neuronal map of a complex mammalian sensory organ, and investigate the mechanisms that influence its composition and maintenance.

To date only a few studies have quantified the number of OSNs that express a given OR (*Bressel et al., 2016*; *Fuss et al., 2007*; *Khan et al., 2011*; *Rodriguez-Gil et al., 2010*; *Royal and Key, 1999*; *Young et al., 2003*). For the scarce data available (<10% of the full repertoire) reproducible differences in abundance have been observed between OSNs expressing different ORs (*Bressel et al., 2016*; *Fuss et al., 2007*; *Khan et al., 2011*; *Young et al., 2003*). This suggests variance in the representation of OSN subtypes exists within an individual, but the extent of variation between individuals is unknown. Moreover, the mechanisms that dictate the abundance of OSN subtypes are poorly understood. Most promoters of OR genes contain binding sites for Olf1/Ebf1 (O/E) and homeodomain (HD) transcription factors (*Young et al., 2011*), and these are involved in determining the probability with which the OR genes are chosen for expression (*Rothman et al., 2005*; *Vassalli et al., 2011*). Enhancer elements also regulate the gene choice frequencies of nearby, but not distally located, ORs (*Khan et al., 2011*). To date, these studies have focused only on a handful of OSN subtypes.

In addition to OR gene choice regulation exerted by genetic elements, it is conceivable that the olfactory system adapts to the environment. The MOE is continually replacing its OSN pool and the birth of every neuron presents an opportunity to shape the proportion of different subpopulations. It is also possible that relative OSN abundances could be altered by regulating the lifespan of each OSN subtype. Indeed activation extends a sensory neuron's life-span (*Santoro and Dulac, 2012*), suggesting that persistent exposure to particular odorants may, over time, increase the relative proportions of the OSNs responsive to them. Some OSN subtypes do reportedly increase in number in response to odor activation, but others do not (*Cadiou et al., 2014*; *Cavallin et al., 2010*;

*Watt et al., 2004*). Whether this variation reflects differences in the biology of OSN subtypes or experimental procedures is unclear.

Here, we fully map OSN diversity in the MOE and characterize the influence of both genetic and environmental factors on its regulation. We show that RNA sequencing (RNAseq) is an accurate proxy for measuring the number of OSNs expressing a particular OR type, and use this approach to quantify, in parallel, the composition of 1115 OSN subtypes in the MOE. We report that, while the repertoire of OSN subtypes is stable across individuals from the same strain, it reproducibly and extensively differs between genetically divergent strains of laboratory mice. We show that under controlled environmental conditions, these stereotypic differences in OSN abundance are directed by genetic variation within regulatory elements of OR genes that predominantly act in cis and are independent of the function of the OR protein. However, we find that persistent, but not continuous, exposure to specific odorants can also subtly alter abundance of the OSN subtypes that are responsive to such stimuli. Taken together, these results show that the OSN repertoire is shaped by both genetic and environmental influences to generate a unique nose for each individual.

## Results

### Olfactory sensory neuron diversity measured by RNAseq

Previously, we characterized the transcriptional profile of the whole olfactory mucosa (WOM) in adult C57BL/6J animals (hereafter termed B6) to generate hundreds of new, extended OR gene annotations (*Ibarra-Soria et al., 2014*). As each OR gene is expressed in only a small fraction of cells within WOM, differences in their abundance are difficult to distinguish from sampling bias. We hypothesized that mapping RNAseq data to significantly extended OR transcripts should increase detection sensitivity. With these models, OR gene mRNA level estimates in adult WOM increase, on average, 2.3-fold, but some increase almost 20-fold (*Figure 1—figure supplement 1A*). Despite this improvement, most OR mRNAs still have relatively low-expression values (*Figure 1A*). Nevertheless, they show a dynamic range of abundance levels (*Figure 1A*, inset) that are consistent between biological replicates, as indicated by their very high correlation values (median rho = 0.89, p<$2.2 \times 10^{-16}$).

To assess whether these low OR mRNA expression values are biologically meaningful or if they represent low-level technical artifacts of RNAseq analysis, we sequenced RNA from WOM of a mouse strain that has a targeted homozygous deletion of the *Olfr7* OR gene cluster on chromosome 9 (*Xie et al., 2000*; *Khan et al., 2011*), and compared their gene expression profile to control mice. From the 94 OR genes of the cluster that have been deleted, 83 (88.3%) have no counts in any of the three biological replicates. The 11 remaining genes have just one or two fragments mapped in only one of the replicates (*Figure 1B*), resulting in normalized counts of less than 0.4. In contrast, the control mice have from 14.2 to 498.1 normalized counts for the same genes. Together these experiments demonstrate that the use of extended gene models significantly increases the sensitivity to detect OR mRNA expression in WOM, and that the full dynamic range of abundances reflects true measures of OR gene expression.

The wide range of stereotypic OR gene expression can be explained by two scenarios, acting alone or in combination (*Figure 1C*): either (1) OR genes with high-expression values are monogenically expressed in more OSNs than those with low-expression values; and/or (2) OR genes are consistently expressed at different levels per OSN. To differentiate between these possibilities, we performed in situ hybridization (ISH) of probes specific to nine OR genes with expression values distributed across the dynamic range. We then counted the number of OSNs in which each OR is expressed (*Figure 1D*). We find very strong correlation between OSN number and RNAseq expression value (rho = 0.98, p=$5 \times 10^{-5}$). We additionally compared OR gene RNAseq expression levels with three independent measures of the number of OSNs expressing the same ORs (*Bressel et al., 2016*; *Fuss et al., 2007*; *Khan et al., 2011*). In all three cases, we find high correlations (*Figure 1—figure supplement 1B–D*). We next collected 63 single mature OSNs from WOM, and determined the OR gene most abundantly expressed in each using a single-cell RNAseq approach (*Saraiva et al., 2016*, unpublished data). If OR expression levels in WOM reflect the proportion of OSNs that express each receptor (*Figure 1C*), the probability of isolating each OSN type is not equal. Indeed, we find a strong selection bias towards OSNs that express OR genes with high RNAseq levels in WOM (hypergeometric test, p=$6.44 \times 10^{-9}$; *Figure 1E*), suggesting those OSN types

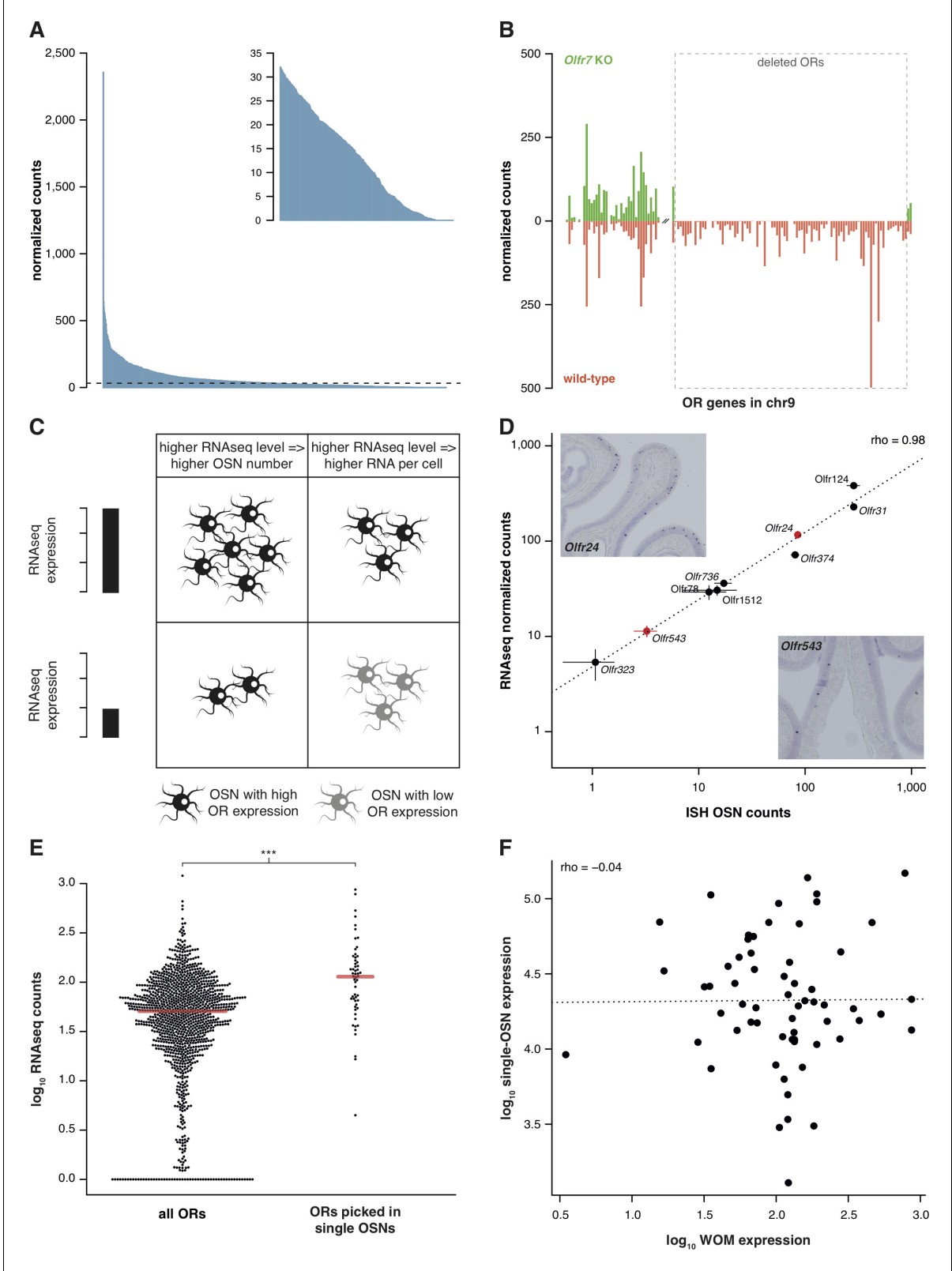

**Figure 1.** RNAseq is highly sensitive for OR mRNA detection and provides a measurement of OSN diversity. (**A**) Barplot of the mean normalized expression of 1249 OR genes from six biological replicates, accounting for gene length. Genes are ordered by decreasing abundance. The horizontal line is the median expression (32.06) and all the genes below it are shown in the inset. (**B**) Mean normalized mRNA expression values for the OR genes in chromosome 9 of the *Olfr7* cluster deletion mouse line (green; n = 3). The corresponding abundances in wild-type animals (orange) are shown as a
*Figure 1 continued on next page*

*Figure 1 continued*

mirror image (n = 3). The break on the x-axis separates the two OR clusters. The dotted box encloses the deleted ORs. (C) Unequal RNAseq expression levels for different OR genes can be explained by two scenarios: (left) an OR gene with high RNAseq levels is expressed by a larger number of OSNs than a gene with low RNAseq abundance; and/or (right) an OR with high RNAseq values is expressed in the same number of OSNs as one with low RNAseq values, but at higher levels per OSN. (D) Comparison of the number of OSNs that express nine OR genes assessed by in situ hybridization (ISH; x-axis) to the corresponding RNAseq values (y-axis). Error bars are the standard error of the mean (ISH n = 4, RNAseq n = 6). The line is the linear regression and the Spearman's correlation coefficient (rho) indicates a very strong correlation. Representative ISH images of two OR genes (in red) are shown. (E) In single-cell RNAseq experiments, 63 OSNs were randomly collected from the MOE. The distribution of OR mRNA expression in WOM samples is plotted (left), alongside the equivalent values for the ORs that were present in the picked single-OSNs (right). There is a significant enrichment (p<6.44 × 10$^{-9}$) toward collecting OSNs that express OR genes with high RNAseq counts in WOM. (F) Comparison of the normalized expression value for the highest OR detected in each of the 63 single-OSNs (y-axis) to the corresponding mean value in WOM (x-axis, n = 3). The line is the linear regression and the Spearman's correlation coefficient (rho) indicates there is no correlation. See also *Figure 1—figure supplement 1*.

The following figure supplement is available for figure 1:

**Figure supplement 1.** RNAseq expression values are a proxy for OSN number.

are more numerous in the olfactory epithelium. Thus, consistent with a recent analysis in zebra fish (*Saraiva et al., 2015*), OR RNAseq values are an accurate measure of the number of each OSN subtype in the mouse WOM (scenario 1). But do consistent differences in OR mRNA levels per cell also contribute (scenario 2)? To test this, we quantified the mRNA levels of the most abundant OR gene in each of the 63 single, mature OSNs, normalized to three stably expressed OSN marker genes (*Khan et al., 2013*). We find OR mRNA levels vary within the single cells, but this does not correlate with expression levels across the WOM (rho = −0.04, p=0.7518) (*Figure 1F*). Analysis of ERCC spike-ins confirmed that the levels of OR mRNAs in single OSNs are reliable. Moreover, the single OSN transcript levels also positively correlate with transcript levels in pools of millions of OSNs (*Saraiva et al., 2016*). Together, these data demonstrate that OR mRNA levels obtained by RNAseq are an accurate proxy for quantifying the diversity of OSN subtypes that express each receptor.

## The OSN repertoire differs between strains of mouse

The relative proportion of each OSN subtype is stable between genetically identical animals. We have previously reported the expression of OR genes in B6 male and female mice (*Ibarra-Soria et al., 2014*). By applying full gene models to these data, here we confirm their OSN distribution profiles are remarkably similar (*Figure 2A*); only 1.2% of the OR gene repertoire is significantly differentially expressed (*Figure 2B*). To investigate whether this OSN distribution is a stereotypic feature of the species, we next reconstructed the WOM transcriptome of a different laboratory strain, 129S5SvEv (hereafter termed 129). The 129 genome has 4.4 million single nucleotide polymorphisms (SNPs) and 0.81 million small indels compared to B6 (*Keane et al., 2011*), of which we find 13,484 SNPs and 1,936 indels within our extended OR gene transcripts. As OR genes are particularly variable in coding sequence between strains of mice (*Logan, 2014*), mapping RNAseq data from other strains to a B6 reference genome results in biases in OR gene expression estimates (*Figure 2—figure supplement 1A*). We therefore generated a pseudo-129 genome on which to map the RNAseq data, by editing the reference genome at all polymorphic sites. We confirmed that the RNAseq expression estimates correlate with the number of OSNs that express the corresponding receptor genes in 129 animals, as judged by in situ hybridization (rho = 1, p=5.5 × 10$^{-6}$; *Figure 2—figure supplement 1B*). From the 1,249 OR genes, we find 462 are significantly differentially expressed (DE) compared to B6 (false discovery rate (FDR) < 5%), representing 37% of the whole repertoire (*Figure 2C,D*).

To determine whether greater genetic diversity influences the variance in OSN repertoire, we repeated this experiment using a wild-derived strain from the *Mus musculus castaneus* subspecies (CAST/EiJ, henceforth CAST). This strain has more than 17.6 million SNPs and 2.7 million indels relative to B6 (*Keane et al., 2011*); of these, we counted that 45,688 SNPs and 6,303 indels are found within our extended OR transcripts. After mapping to a pseudo-CAST genome (*Figure 2—figure supplement 1C*), 634 OR genes are significantly differentially expressed (FDR < 5%) compared to B6, constituting 50.8% of the whole OR repertoire (*Figure 2E,F*). The changes in expression for

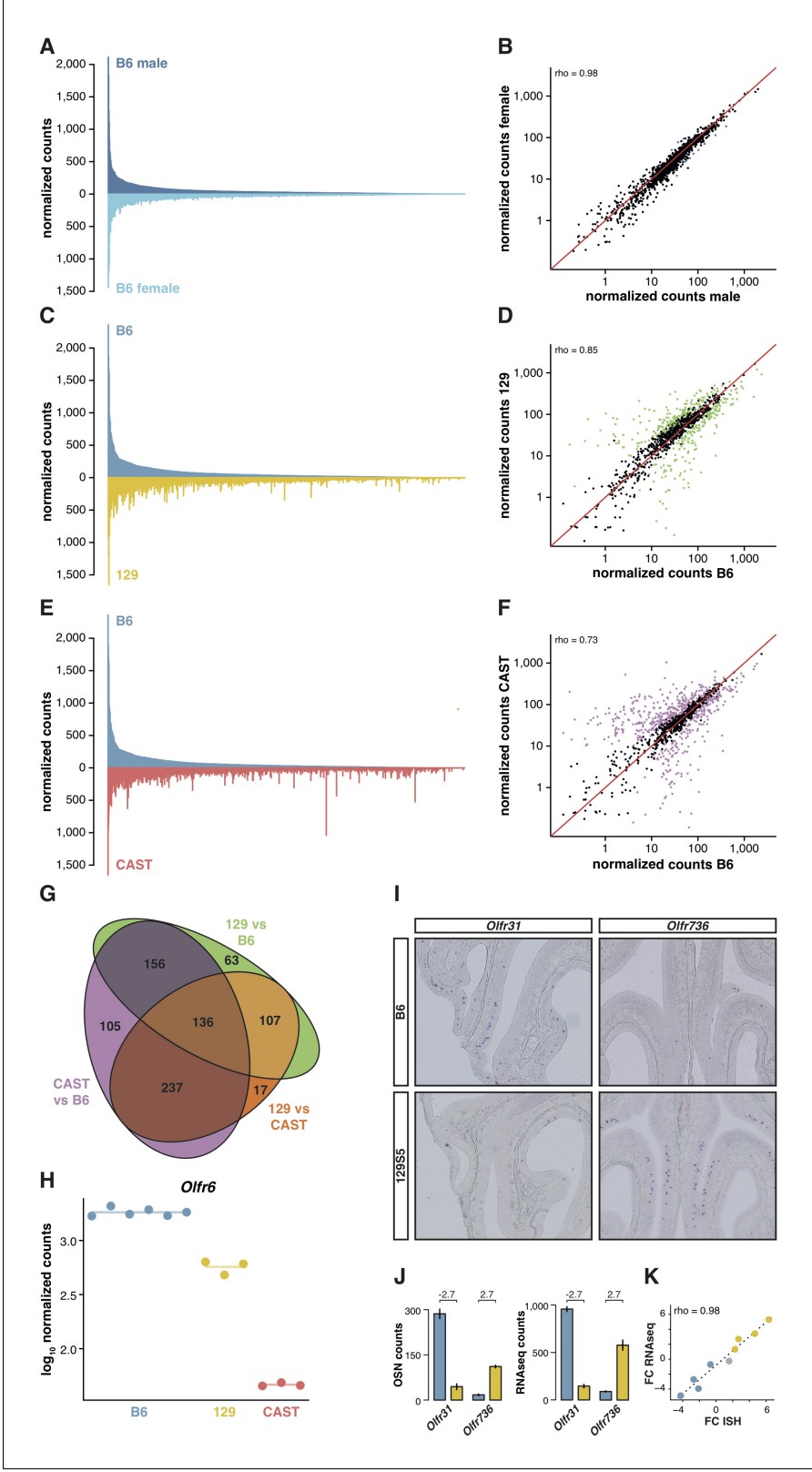

**Figure 2.** OSN diversity varies between mouse strains. (**A**) Mirrored barplot of the mean normalized RNAseq expression values for the OR genes in male (dark blue, top) and female (light blue, bottom) B6 animals (n = 3). (**B**) Scatter plot for the same data, with the Spearman's correlation (rho) indicating a strong correlation. The red line is the 1:1 diagonal. Significantly differentially expressed (DE) OR genes are represented in blue, non-DE genes are in

*Figure 2 continued on next page*

*Figure 2 continued*

black. (**C**) Same as in (**A**) but with the average B6 expression in blue (both males and females, n = 6) compared to the corresponding 129 expression values in yellow (n = 3). (**D**) Corresponding scatter plot, with the significant DE genes in green. (**E**) Same as in (**A**) but comparing the B6 expression in blue (n = 6) to the CAST abundances in red (n = 3). (**F**) Corresponding scatter plot, with DE genes in purple. (**G**) Venn diagram illustrating the intersection of DE OR genes between the pairwise comparisons of the three strains. (**H**) An example of an OR gene, *Olfr6*, that is DE in all strain comparisons. (**I**) Representative in situ hybridizations (ISH) on coronal slices of B6 and 129 MOEs for two OR genes, *Olfr31* and *Olfr736*, that are DE between these strains. (**J**) The quantification of OSNs expressing each OR gene in B6 (blue) and 129 (yellow) are plotted alongside the corresponding RNAseq counts. The $\log_2$ fold-changes between the strains are indicated. (**K**) Fold-change between the strains obtained from ISH data (x-axis) or RNAseq counts (y-axis) for four DE OR genes expressed higher in B6 (blue), four expressed higher in 129 (yellow) and one expressed at equivalent levels in both strains (grey); these include *Olfr31* and *Olfr736*. The line is the linear regression and the Spearman's coefficient (rho) indicates a strong correlation between OSN and RNAseq counts. See also *Figure 2—figure supplement 1*.

The following source data and figure supplement are available for figure 2:

**Source data 1.** OR expression data in three strains of mice.
**Source data 2.** Novel OR alleles in the CAST genome.
**Figure supplement 1.** Differences in genetic background result in great variance in OSN subtype diversity.

---

some OR genes are dramatic: 132 genes have differences of at least eight fold. Taking all pairwise comparisons into account (including 129 vs CAST, *Figure 2—figure supplement 1D,E*), 821 OR genes (65.7%) are DE between at least two strains. One hundred and thirty-six of these are DE in all three pairwise comparisons (*Figure 2G*); for example, there are consistently different numbers of *Olfr6*-expressing OSNs in each strain (*Figure 2H*).

To determine if the DE OR genes between strains reflect differences in the proportions of OSN subtypes, we performed ISH with probes specific to OR genes with significantly different expression values between B6 and 129 (*Figure 2I*). We then counted the number of OSNs that express nine different OR mRNAs, in each strain, and compared this with their RNAseq expression values (*Figure 2J*). We find a high correlation between the difference in OSN number and the difference in RNAseq expression values between B6 and 129 (rho = 0.98, p=5 $\times$ 10$^{-5}$; *Figure 2K*), demonstrating our RNAseq-based approach accurately measures the difference in OSN repertoires between strains.

OR gene clusters are enriched in copy number variants (CNVs) between individual human (*Nozawa et al., 2007*; *Young et al., 2008*) and mouse strain genomes (*Graubert et al., 2007*). Thus, it is possible that variance in OSN subtype representations are a consequence of different numbers of highly similar OR genes between strains. To assess this, we mined CAST genome sequence data (*Keane et al., 2011*) for heterozygous SNPs within annotated OR genes. We identified 51 ORs that contain 10 or more heterozygous SNPs, an indication of additional alleles. Using genome sequencing data from these genes, we identified 30 novel or misassembled OR genes. We remapped the CAST RNAseq data to a pseudo-CAST genome incorporating these new OR alleles and re-estimated the expression of the OR repertoire. The overall abundance profile remains unchanged except for 36 genes (*Figure 2—figure supplement 1F*). To assess whether this accounts for the observed differential expression between strains, we compared these estimates to B6. Only 12 of 634 OR genes lose their DE status, and 4 OR genes now become DE (*Figure 2—figure supplement 1G*). Thus, while differences in OR gene copy number minimally contribute to the diversity in OSN repertoire between three strains of mice, other mechanisms are responsible for most of the variation.

## Genetic background instructs OSN diversity independent of odor environment

Genetically divergent mouse strains produce different chemical odortypes in their urine (*Kwak et al., 2012*; *Yamaguchi et al., 1981*) and amniotic fluid (*Logan et al., 2012*). Therefore,

each strain of mouse, when housed in homogeneous groups, is exposed to a unique pre- and post-natal olfactory environment. As odor exposure alters the life-span of OSNs in an activity-dependent manner (*François et al., 2013*; *Santoro and Dulac, 2012*; *Watt et al., 2004*), genetic variation could regulate OSN population dynamics either directly or indirectly, via odortype. We, therefore, devised an experiment to test and differentiate the influence of the olfactory environment from the genetic background.

We transferred four to eight-cell stage B6 and 129 zygotes to F1 mothers to ensure they experienced an identical in utero environment. At birth, B6 litters were cross-fostered to B6 mothers and 129 litters to 129 mothers. In addition, B6 litters received a single 129 pup, and 129 litters a single B6 pup. Therefore, each litter experienced a characteristic olfactory environment, but one animal (the *alien*) had a different genetic background from the others (*Figure 3A*). At 10 weeks of age, we quantified the OSN repertoires of six alien animals and six cage-mates using RNAseq. We found that the OSN repertoires cluster in two groups, clearly defined by genetic background (*Figure 3B*). The correlation coefficient for any two B6 samples was on average 0.97, with no significant difference between the environments (t-test, p=0.09). In contrast, the correlation for any B6 with a 129 sample had a mean of 0.89, which is significantly lower (t-test, p=$3.8 \times 10^{-12}$). Five hundred and seven OR genes, among 5475 other genes are DE between these mice when grouped by strain (*Figure 3—figure supplement 1A*). In striking contrast, across the whole transcriptome we find only mRNA from two genes that show differences in expression according to odor environment, both of which encode ORs (*Figure 3C*, *Figure 3—figure supplement 1B*). These data demonstrate that the diversity in OSN repertoire we observe between strains is almost entirely dictated by direct genetic effects. In a controlled environment, the influence of odortype on the development and maintenance of the MOE is minimal, perhaps restricted to only a few OSN subtypes.

## OSN diversity profiles are independent of OR function and are controlled in *cis*

The indifference of the OSN repertoire to the olfactory environment suggests its development and maintenance is not influenced by the specific activity of OR proteins or, by inference, their protein coding sequence. To further test this, we analyzed the OSN repertoire of newborn pups. We identify the presence of 1,198 (95.9%) OSN subtypes across a dynamic range of abundance (*Figure 4A*). The differential proportions of OSNs expressing particular OR genes are therefore already present during the development of the MOE, suggesting that it is not dependent on the activity of the OSNs nor on differences in OSN life-span.

Next, we analyzed the expression of ORs that are pseudogenized and do not produce receptor proteins capable of odor-mediated activity, but can be co-expressed with functional ORs (*Serizawa et al., 2003*; *Saraiva et al., 2016*). These are represented in OSNs with a very similar distribution to functional OR genes (*Figure 4B*). Moreover, we analyzed the OR genes that encode identical protein-coding sequences between different strains. 36.3% of the OSN subtypes that express identical ORs are differentially represented between 129 and B6. 44.8% are differentially represented between CAST and B6. Together, these results suggest that the proportion of each OSN subtype is not dependent on its endogenous OR receptor activity.

To directly test whether the abundance of a particular OSN subtype is influenced by the identity of the receptor protein it expresses, we used CRISPR-Cas9 to replace only the coding sequence of *Olfr1507* with that of *Olfr2* (referred to as *Olfr2 > Olfr1507*), in a pure B6 genetic background (*Figure 4C*). OSNs expressing *Olfr1507* are the most common subtype in B6, while *Olfr2* expressing OSNs are ranked 334th by decreasing abundance. Homozygous *Olfr2 > Olfr1507* animals have 47 fold more *Olfr2*-expressing OSNs compared to controls, and is the highest subtype in these animals (*Figure 4D*). DE analysis of OR genes supports the striking reciprocal differences in *Olfr1507* and *Olfr2*-expressing OSNs in the *Olfr2 > Olfr1507* animals, but we also find 118 other OSN subtypes with significant, albeit comparatively subtle, differences (over 90% have fold-changes < 2) (*Figure 4E*). Taken together, these data indicate that the extensive variance in OSN subtype composition we observe in mice is determined by the wider genetic architecture of the animal, and is independent of the function of the OR protein each subtype expresses.

To investigate how genetic background influences OSN subtype abundances, we mined 129 and CAST whole genome sequences (*Keane et al., 2011*) for SNPs and short indels in regulatory regions of OR genes. We find that differentially represented OSN subtypes express OR genes with

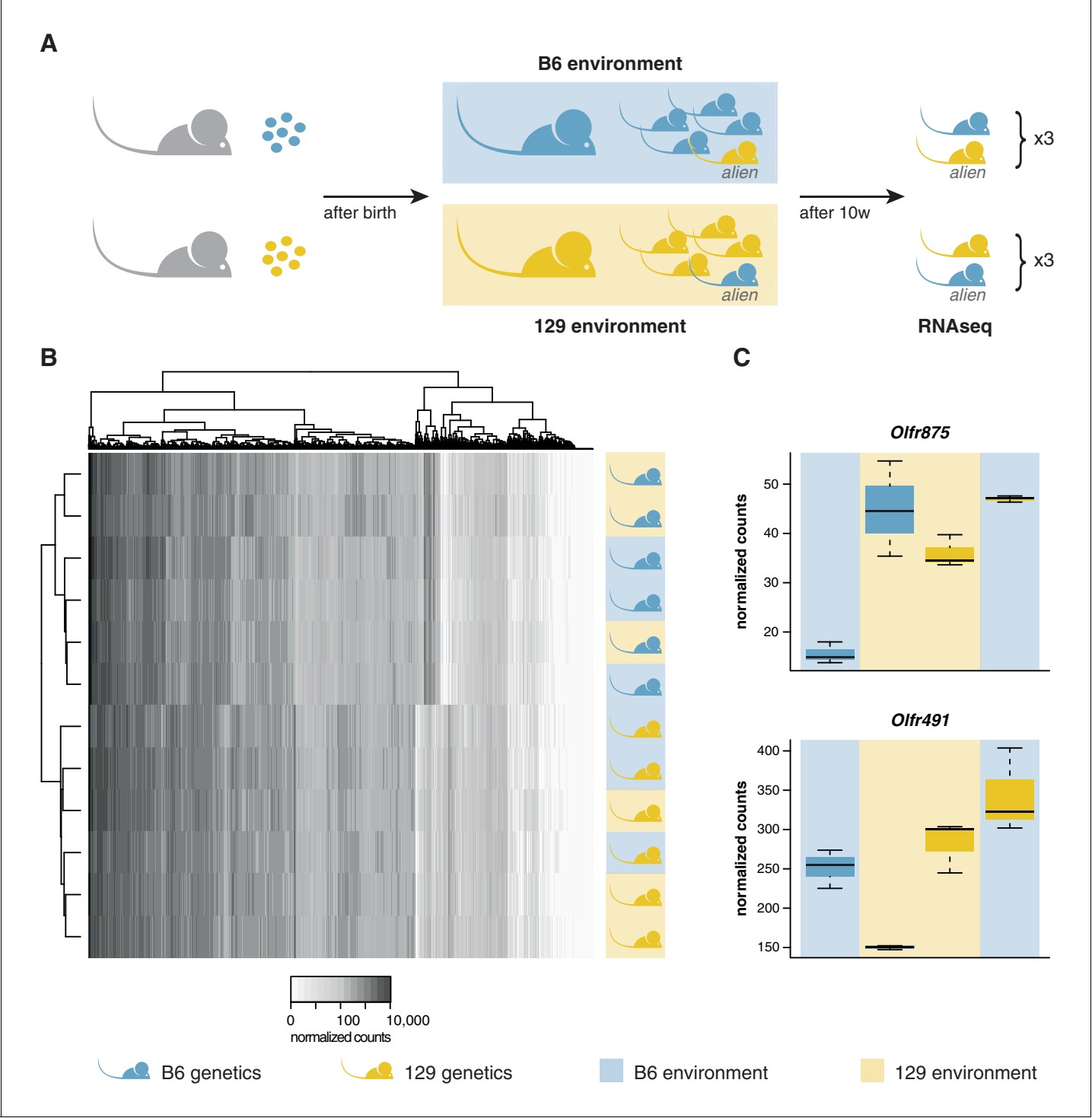

**Figure 3.** OSN diversity is determined by the genetic background and not by the olfactory environment. (A) Experimental strategy to differentiate genetic from environmental influences on OSN diversity. B6 (blue) and 129 (yellow) embryos, depicted as circles, were transferred into F1 recipient mothers (grey). After birth, the litters were cross-fostered to B6 and 129 mothers, respectively. Each B6 litter received one 129 pup (the alien) and vice versa. After 10 weeks, the WOM was collected for RNAseq from three aliens from each strain, and one cage-mate each. (B) Heatmap of the expression of the OR genes (columns) in all 12 sequenced animals (rows). Samples cluster by the genetic background of the animals. The strain and environment of each mouse is indicated through shading (right). (C) Differential expression analyses revealed mRNA from only two genes, *Olfr875* and *Olfr491*, that are significantly altered based on the olfactory environment. Expression values are shown for each group. Blue and yellow boxes indicate B6 or 129 animals respectively, and the background indicates the olfactory environment. See also *Figure 3—figure supplement 1*.

*Figure 3 continued on next page*

*Figure 3 continued*

The following figure supplement is available for figure 3:

**Figure supplement 1.** Genetic but not environmental factors regulate OSN subtype abundance.

significantly greater amounts of variation in their coding sequence, whole transcript and regions of 300 bp or 1 kb upstream of the transcription start site, for both the 129 and CAST genomes (Mann-Whitney one tail, $p<0.02$ for 129 and $p<0.0002$ for CAST; *Figure 4—figure supplement 1A*). Further, we scanned OR gene promoters for O/E and HD binding sites. In the CAST genome, 58 and 310 putative OR promoters have gains or losses of O/E and HD-binding sites respectively, compared to the B6 genome. In contrast, only 12 and 46 OR promoters show differences in the number of O/E and HD-binding sites, respectively, when comparing the 129 and B6 genomes.

We therefore hypothesized that OSN subtype repertoires are generated via sequence variance in OR gene promoter and/or local enhancer elements, which dictate the frequency of OR gene choice. For two OR gene clusters, it has been demonstrated that enhancer/promoter interactions act in cis and do not influence the expression of the homologous OR allele on the other chromosome (*Fuss et al., 2007*; *Khan et al., 2011*; *Nishizumi et al., 2007*). However, recent chromosome conformation capture experiments revealed interchromosomal interactions between OR enhancer elements (*Markenscoff-Papadimitriou et al., 2014*). Moreover, the differential representation of 118 other OSN subtypes in the *Olfr2 > Olfr1507* line (*Figure 4E*), 108 of which express ORs that are located on a different chromosome from *Olfr1507*, is consistent with the possibility that genetic modification of one OR locus directly influences the probability of choice in other ORs, in trans.

To determine whether the genetic elements that instruct the whole OSN repertoire are *cis-* or *trans*-acting, we carried out an analysis at the OR allele level in B6 × CAST F1 hybrids. Following the logic of (*Goncalves et al., 2012*), if the genetic elements act in cis then we would expect the OSN subtypes that differ between B6 and CAST to be maintained between OSNs expressing the corresponding B6 and CAST alleles within an F1 hybrid. On the other hand, if the elements act in trans the number of OSNs that express the B6 derived allele in the F1 would not differ from those that express the CAST allele.

Within F1 mice, 840 OSN subtypes (67.2%) expressed OR mRNAs that could be distinguished at the allelic level. The ratios between B6 and CAST OSN subtype abundance (F0) strongly correlate with the ratios between alleles in the F1 hybrids at approximately 1:1 (concordance correlation coefficient (ccc) = 0.86; *Figure 4F*). In other words, taken across over 800 OSN subtypes, those expressing a B6 OR allele in F1 animals have the same repertoire as the B6 parent, while the subtypes expressing the CAST OR allele match the CAST parent (*Figure 4G*). To better understand the significance of any deviations from 1:1 concordance in specific OSN subtypes, we corrected for technical noise associated with distinguishing allelic expression at low counts (see Materials and methods; *Figure 4—figure supplement 1B,C*). The corrected data has a stronger correlation (ccc = 0.92) and the best fit of this data matches 1:1 concordance ($C_b = 0.999$). We therefore conclude that, collectively, the genetic elements dictating the abundance of over 800 OSN subtypes act in cis. However, we cannot exclude the possibility that a small contribution from *trans*-acting elements may account for subtle deviations from unity for some OSN subtypes.

Taken together, these data are consistent with a predominant model where genetic variation in local, non-coding regulatory elements determines the probability with which each OR gene is chosen early in OSN neurogenesis.

## Acute but not chronic odor exposure affects OR mRNA expression in the WOM

Previous studies have shown that OSNs activated by their cognate ligands have increased life-span (*François et al., 2013*; *Santoro and Dulac, 2012*; *Watt et al., 2004*). With time, longer survival rates should translate into enrichment in the neuronal population, compared to those OSN types that are mostly inactive (*Santoro and Dulac, 2012*). However, we found no evidence of different strain- or sex-derived odors influencing the OSN repertoire (*Figures 2A* and *3B*). Because these odor exposures were temporally constant, we hypothesized that the absence of an observed environmental

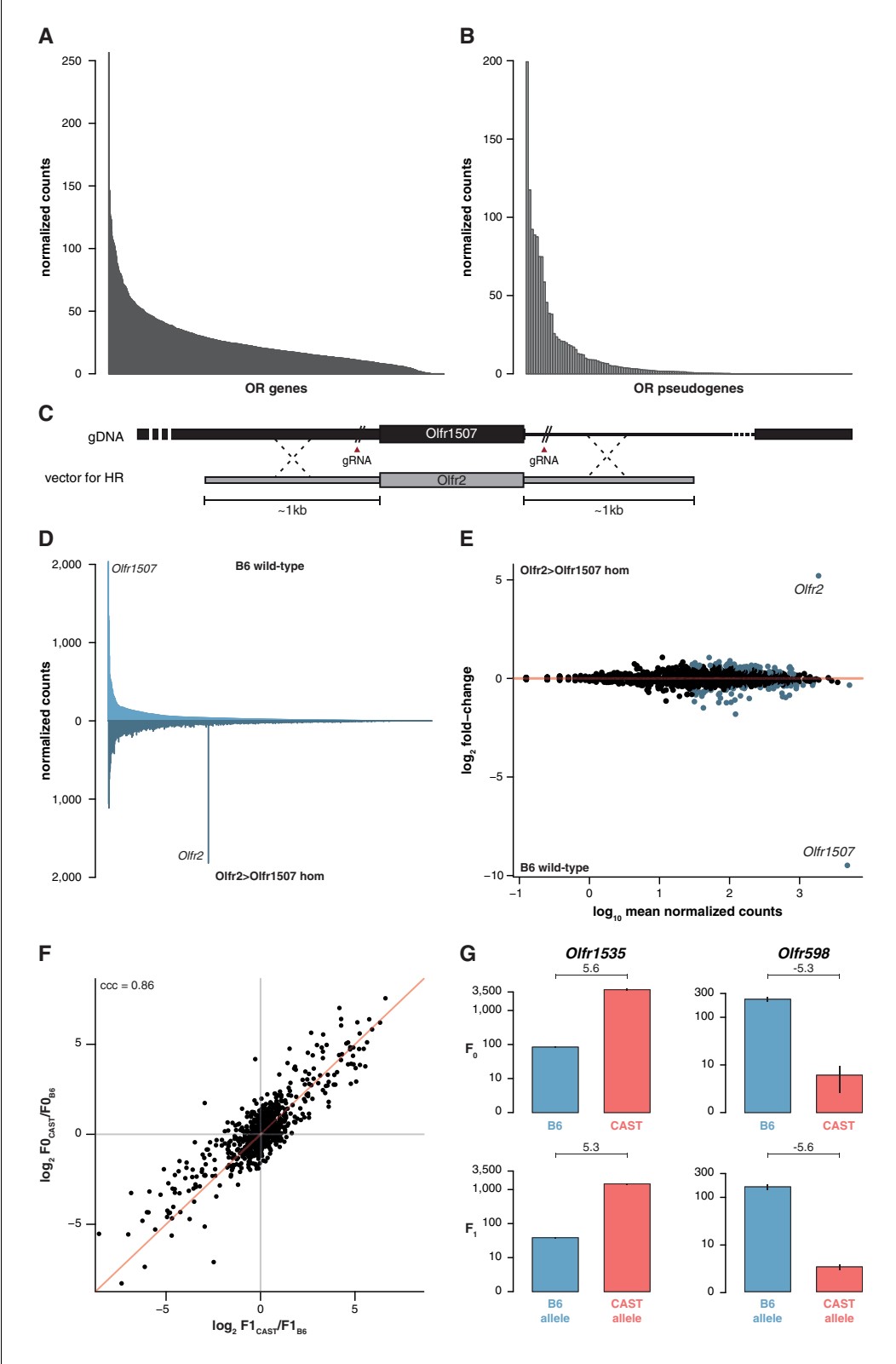

**Figure 4.** OSN diversity is independent of OR activity and is controlled in *cis*. (A) Mean normalized expression of the OR mRNA in the WOM of newborn B6 animals, arranged from most to least abundant (n = 3). (B) Mean normalized mRNA expression of 134 annotated OR pseudogenes in the B6 adult WOM (n = 6). (C) A genetically modified mouse line was produced that contains the coding sequence (CDS) of *Olfr2* in place of *Olfr1507* (*Olfr2 > Olfr1507*). The strategy combined the use of CRISPR-Cas9 technology to create double-strand breaks on either side of the *Olfr1507* CDS, and a

*Figure 4 continued on next page*

*Figure 4 continued*

DNA vector containing the *Olfr2* CDS along with ~1 kb homology arms for homologous recombination (HR). (D) Mirrored barplot of the mean normalized mRNA expression values for the OR repertoire in B6 animals (light blue, top; n = 4) and in *Olfr2 > Olfr1507* homozygous (hom) mutants (dark blue, bottom; n = 4). *Olfr2* becomes the most abundant OR and *Olfr1507* is no longer expressed in the genetically modified line. (E) Scatter plot of the mean normalized counts (x-axis) of OR genes versus the $\log_2$ fold-change between *Olfr2 > Olfr1507* homozygotes and WT controls (y-axis, n = 4). OR genes that are significantly differentially expressed are represented in blue. *Olfr2* and *Olfr1507* are strikingly different, whereas the rest of the repertoire is equivalent or very slightly altered. (F) Comparison of the fold-change of the CAST versus B6 OR expression (y-axis) to the fold-change between the CAST and B6 alleles in the F1 (x-axis). The genes fall largely on the 1:1 diagonal (red line) indicating the mRNA expression pattern observed in the parents is preserved in the F1 and thus OR abundance is controlled in *cis*. The concordance correlation coefficient (ccc) is indicated, which quantifies the correlation between the two fold-change estimates while correction for agreement on the x=y line. (G) Examples of the normalized mRNA expression in the parental strains (top) of an OR gene that is more abundant in CAST (*Olfr1535*) or in B6 (*Olfr598*). The corresponding mRNA abundance of each allele in the F1 (bottom) is preserved. The $\log_2$ fold-change is indicated for each comparison. Error bars are the standard error of the mean. See also *Figure 4—figure supplement 1*.

The following source data and figure supplement are available for figure 4:

**Source data 1.** OR expression data in the *Olfr2 > Olfr1507* mouse.
**Source data 2.** Number and coordinates of OE and HD binding sites in OR gene promoters.
**Figure supplement 1.** Differentially represented ORs have more variation.

influence on OSN repertoire could be due to olfactory adaptation (a reduction of specific olfactory sensitivity due to prolonged odor exposure, reviewed in [*Zufall and Leinders-Zufall, 2000*]).

To test this, we exposed mice to a mix of four chemically distinct odorants (acetophenone, eugenol, heptanal and (R)-carvone). The odorant mixture was added to the drinking water supplied to the animals to avoid adaptation, such that they could smell the odor mixture when they approached the bottle to drink (*Figure 5A*). We collected the WOM from animals exposed to the odorants for 24 weeks from birth, along with water-exposed controls, and performed RNAseq. DE analysis reveals 36 OR genes with significantly different mRNA levels (FDR < 5%), with similar numbers more or less abundant in the exposed animals (*Figure 5B*, *Figure 5—figure supplement 1*). We selected seven OR genes with the biggest fold-changes in mRNA level for which specific TaqMan qPCR probes were available, and validated their expression levels in a larger cohort. The results indicate that all the tested genes have mRNA levels that are statistically significantly different from controls (t-test, FDR < 5%) and the direction of the expression changes are concordant with the RNAseq data (*Figure 5C*).

To characterize the temporal dynamics of these OR mRNAs, we tested their expression after different periods of exposure (1, 4 and 10 weeks) in independent samples. After 1 week of treatment, none showed significant differences from controls, which is expected since young pups do not drink from the odorized water bottle. After 4 weeks, three of the OR genes are DE from controls, and at 10 weeks, five out of the seven receptors are DE (t-test, FDR < 5%; *Figure 5C*). To assess the plasticity of these changes, we stimulated a group of animals for four weeks, and then removed the odor stimuli for an additional 6 weeks. In these mice, none of the OR genes are DE from controls (*Figure 5D*). Thus, the abundance of specific OR types in WOM is increasingly altered, over a period of weeks to months, upon frequent environmental exposure to defined olfactory cues. These differences are reversible and require persistent stimulation to be maintained.

To investigate whether olfactory adaptation blocks this effect, we presented the same odor mixture on a cotton ball inside a tea strainer (*Figure 5E*), such that the stimuli are present in a sustained manner. None of the same seven OR genes are DE after 24 weeks, nor are any consistently dysregulated during the course of the exposure experiment (t-test, FDR < 5%; *Figure 5E*). Therefore, when odorants are present in the environment in a constant manner (similar to those differentially produced by gender or strains of mice), the OR mRNA abundance levels most responsive to acute exposure remain unchanged.

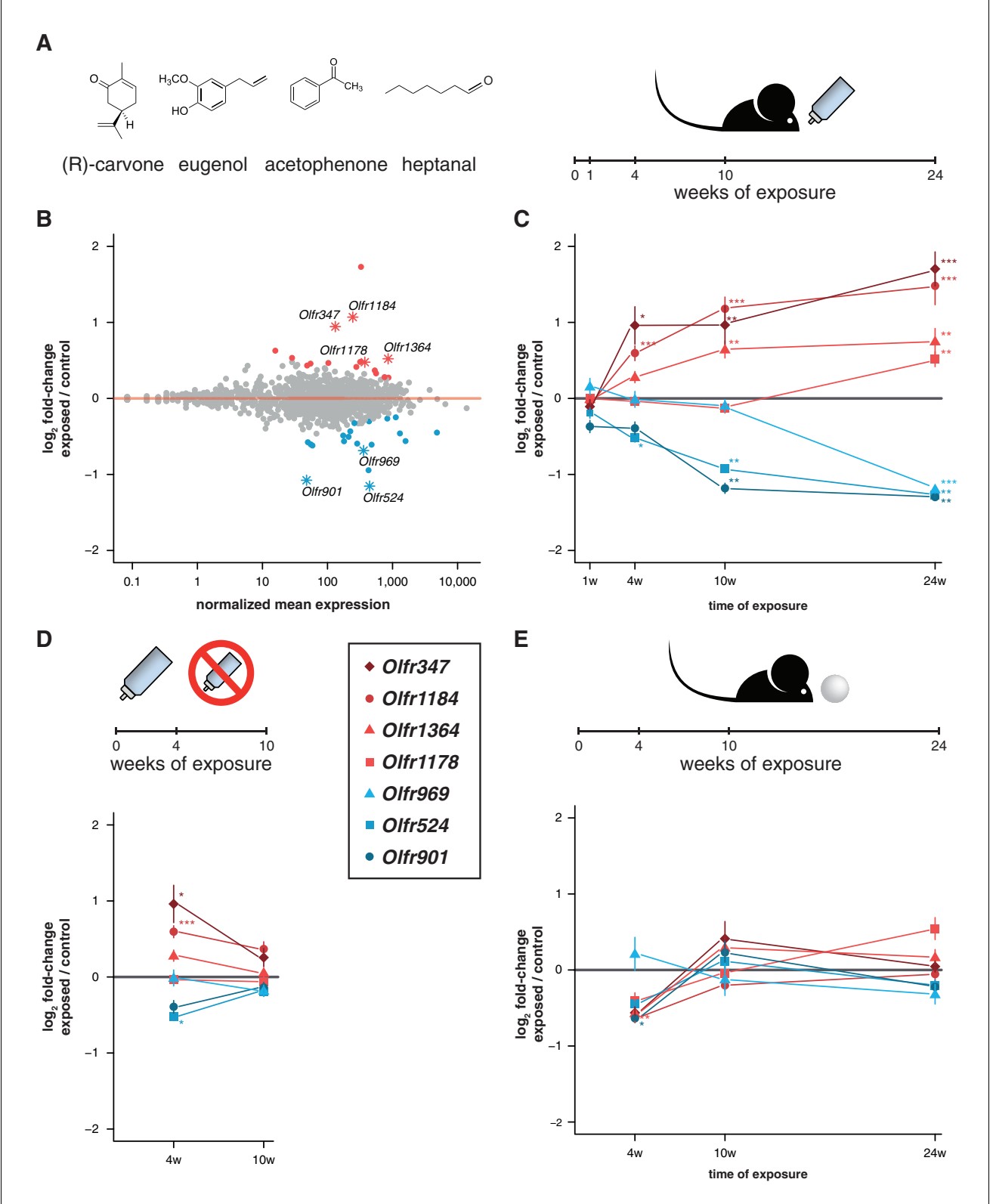

**Figure 5.** Acute but not chronic exposure to odors alters OR mRNA abundance. (**A**) Four different odorants were mixed together and used to stimulate B6 animals. In an acute paradigm, the odor mix was added to the drinking water supplied to the animals and WOM was collected at different time-points. (**B**) WOM from animals exposed for 24 weeks and matched controls were sequenced (n = 6). The plot shows the normalized mean mRNA expression value (x-axis) for each OR gene compared to its fold-change in exposed versus control samples (y-axis). Genes highlighted in red or blue

*Figure 5 continued on next page*

*Figure 5 continued*

have significantly up- or downregulated mRNAs, respectively. OR genes represented by an asterisk were selected for further validation. (**C**) qRT-PCR validation of the DE genes highlighted in (**B**). The mean fold-change between exposed and control samples is plotted for animals exposed for differing periods of time (x-axis). After 24 weeks of exposure, all the genes are significantly DE (n = 8–13). (**D**) Animals were acutely exposed to the odor mix for four weeks and then the stimulus was removed for 6 weeks. After the recovery period none of the OR mRNAs are significantly different from controls (n = 8). (**E**) A chronic exposure paradigm was tested by presenting the odor mix on a cotton ball, placed in the cages of the animals for 24 hr a day. The WOM was collected at different time-points. The genes previously shown to be DE were tested by qRT-PCR and none show consistent changes in mRNA levels across time (n = 3–10). T-test, FDR < 5%; * < 0.05, ** < 0.01, *** < 0.001. Error bars are the standard error of the mean. See also *Figure 5—figure supplement 1*.

The following source data and figure supplement are available for figure 5:

**Source data 1.** OR expression data in odor-exposed mice.

**Figure supplement 1.** Specific OSN subtypes change in abundance upon olfactory stimulation.

## Differential regulation of OR gene mRNAs is odorant-specific

If temporal differences in OR mRNA abundance are a consequence of odorant-specific activity, exposure to different odorants should lead to the differential expression of discrete subsets of OR genes. To test this, we odorized the drinking water with (R)-carvone alone, heptanal alone, or with the combination of both (*Figure 6A*). After 10 weeks of exposure, we tested the expression of the seven DE OR mRNAs that were responsive to the four odor mix (acetophenone, eugenol, heptanal and (R)-carvone), by TaqMan qRT-PCR. None of the genes are significantly DE in the animals exposed to (R)-carvone alone. However, four of the seven OR genes have mRNA levels significantly different in the animals exposed to heptanal, or to the combination of both odorants (t-test, FDR < 5%; *Figure 6—figure supplement 1A*). We next carried out a transcriptome-wide analysis by RNAseq, finding 43 OR genes significantly DE in at least one of the conditions (FDR < 5%) of which 32 (74.4%) are upregulated in the odor-stimulated animals (*Figure 6B*). Exposure to (R)-carvone or heptanal resulted in a change in mRNA expression of 15 and 20 OR genes, respectively. These sets of receptors are almost completely independent, with only one OR mRNA significantly upregulated in both groups (*Olfr538*; *Figure 6C–D*). The animals that were exposed to both odorants simultaneously showed significant changes in mRNA levels for 24 OR genes, 15 of which are shared with the individually exposed groups (hypergeometric test, p=$1.87 \times 10^{-19}$). Almost 40% of the ORs that show significant changes when exposed to all four odorants (*Figure 5B*) are also significantly altered in one or more of the groups exposed to (R)-carvone, heptanal or their combination. Thus, together these data demonstrate that acute environmental exposure to the odorants alters the global expression of around 1.2–1.6% of OR genes in the WOM. These changes are odor-specific and reproducible in isolation and in increasingly complex mixtures.

To investigate whether DE OR genes are directly activated by the environmental odorants, we expressed a subset (*Olfr538*, *Olfr902*, *Olfr916*, *Olfr1182*, *Olfr347* and *Olfr524*) in a heterologous system (*Zhuang and Matsunami, 2008*) and challenged them with increasing concentrations of (R)-carvone and heptanal. Half of the DE ORs we tested were responsive in vitro (*Figure 6E–F*, *Figure 6—figure supplement 1B*): for example, *Olfr538* displayed a dose-dependent response to (R)-carvone (*Figure 6E*) and *Olfr524* was responsive to heptanal (*Figure 6F*).

Some odorants, including heptanal, are known to be decomposed by enzymes present in the nasal mucus (*Nagashima and Touhara, 2010*) such that in vivo exposure to an odorant may result in stimulation of the OSNs with chemically distinct byproducts. We therefore employed a recently published deorphanization system to identify the ORs that respond to heptanal stimulation in vivo (*Jiang et al., 2015*). This strategy exploits the phosphorylation of the S6 ribosomal subunit when an OSN is activated. Thus, by coupling pS6-immunoprecipitation (ps6-IP) and RNAseq, the OR mRNAs expressed in the activated OSNs can be identified. We exposed mice to two concentrations of heptanal for an hour, and sequenced the mRNAs from OSNs after pS6-IP. Twelve and 210 OR mRNAs were significantly enriched (FDR < 5%) upon exposure to 1% and 100% heptanal, respectively, compared to controls. Over half of the DE ORs after 10-week acute exposure to heptanal (*Figure 6B*) are also DE in the pS6$^+$ cells (11 out of 20; *Figure 6G*), which is significantly more than expected by

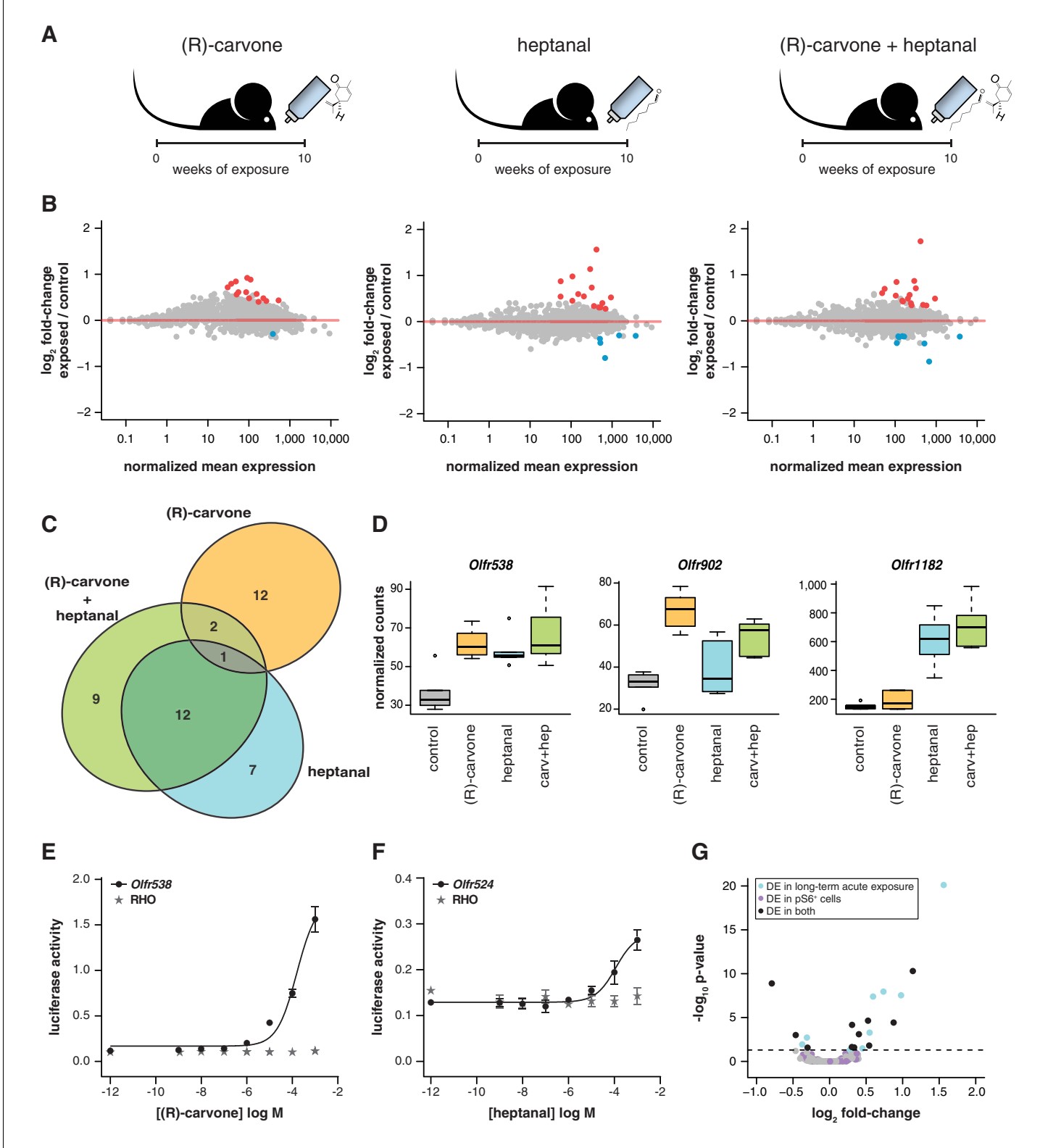

**Figure 6.** Odor-mediated changes in OR mRNA abundance are receptor specific. (**A**) B6 animals were acutely exposed for 10 weeks to (R)-carvone, heptanal or both. (**B**) The fold-change of exposed compared to control animals based on RNAseq data (y-axis) is plotted against the OR genes mean mRNA abundance (x-axis), for each of the experimental groups (n = 6). Genes in red or blue have significantly up or downregulated mRNAs, respectively. (**C**) Venn diagram showing the intersections of the DE OR genes in each of the exposure groups. Only one OR mRNA changed in all

*Figure 6 continued on next page*

*Figure 6 continued*

groups; all the other are specifically altered upon exposure to (R)-carvone or heptanal. (D) Examples of an OR mRNA that changes in all groups (*Olfr538*), one that is specific to stimulation with (R)-carvone (*Olfr902*) and one that responds only to heptanal (*Olfr1182*). (E) Dose-response curve for HEK293 cells expressing Olfr538 (black) and challenged with increasing concentrations of (R)-carvone. HEK293 cells expressing a RHO-tag only (grey) were challenged with the same concentrations of (R)-carvone as a control. (F) Dose-response curve for cells expressing Olfr524 (black) and challenged with heptanal, control cell responses are represented in grey. Error bars are the standard error of the mean. (G) Comparison of DE genes identified after 10 weeks of acute exposure to heptanal to those found via an in vivo deorphanization strategy. On the x-axis is the fold change of acutely exposed versus control animals with the corresponding p-value on the y-axis. The horizontal line represents the cutoff for significance. Each dot is an OR gene; those called significantly DE in both assays are shown in black, while those responding in only one experiment are in blue and purple. Half (11/20) of all the DE genes in the acute exposure experiment are identified in the deorphanization assay, suggesting that the changes are indeed mediated by OSN activation by heptanal. See also *Figure 6—figure supplement 1*.

The following source data and figure supplement are available for figure 6:

**Source data 1.** OR expression data in odor-exposed mice.
**Figure supplement 1.** Olfactory-induced changes in OSN abundance are odor-specific.

chance (hypergeometric test, p=0.0001). Thus, using both in vitro and in vivo methods, we conclude that long-term odor-mediated changes in OR gene expression occurs via direct activation of OSNs expressing those receptors.

## Discussion

We have exploited the power of RNA sequencing and the monogenic and monoallelic nature of OR gene expression, to comprehensively characterize the full neuronal diversity of a mammalian nose. We have discovered that the representation of OSN subtypes within an individual is highly unequal but stereotyped between animals of the same genetic background. We show that non-coding genetic variation results in high divergence of the relative proportions of different OSN subtypes, with most being susceptible to altered abundance. OSN diversity is predominantly controlled by genetic elements that act in cis and the abundance of a given OSN subtype is not affected by the sequence or function of the receptor it expresses, in a sustained olfactory environment. However, the persistent but interleaved presentation of olfactory stimuli can alter the representation of specific ORs in an activity-dependent manner, thus subtly shaping the genetically-encoded neuronal repertoire to the olfactory environment.

### The MOE is a genetically determined mosaic of OSN subtypes

The process of OR gene choice, stabilization and exclusion during OSN maturation is poorly understood. Occasionally, it is referred to as a *random* process (*McClintock, 2010*; *Rodriguez, 2013*), suggesting there is no pattern or predictability to the outcome. However, our data indicates that, at the OSN population level, the result of this process is deterministic. A particular genetic background in controlled environmental conditions reproducibly generates an OSN population with fixed, unequal proportions of the different OSN subtypes. Thus, the process that generates this profile is more accurately described as *stochastic*. Despite divergence in the profiles generated by different genomes, all show a similarly shaped distribution: a small proportion of OSN subtypes are present at high levels with a rapid decay in abundance thereafter. In fact, 3.6% or less of the OSN subtypes contribute to 25% of the overall neuronal content of the WOM. We find that unequal OSN distributions are already present at birth (*Figure 4A*), suggesting the genetic influence is on the process of OR gene choice/stabilization rather than modulating neuronal survival.

Proximity to the H element, a cluster-specific enhancer, increases the frequency in which an OR is represented within the OSN population (*Khan et al., 2011*). Therefore, the most highly represented OSN types may express OR genes located close to other strong enhancers. However, we propose that genetic variation in enhancers is not sufficient to account for the full diversity of differences in OSN subtypes between strains, as different ORs located adjacent to one another within a cluster are frequently represented very differently. Recently, it has been proposed that higher levels of OR

transcription per cell may result in more OSNs expressing that receptor due to increased success in a post-selection refinement process (*Abdus-Saboor et al., 2016*). Our measurements of OR mRNA expression levels in 63 single OSNs (*Figure 1F*) do not support this hypothesis. Instead, our data are consistent with a model where non-uniform probabilities of OR choice are instructed by genetic variation in both specific OR promoters and enhancers. Supporting this model, we identified many putative promoters for differentially represented ORs where genetic variation has altered the number of Olf1/Ebf1 (O/E) and homeodomain (HD) transcription factors binding sites between the mouse strains, both sequences known to influence the probability of OR choice (*D'Hulst et al., 2016*; *Vassalli et al., 2011*). Moreover, through analysis of F1 hybrids, we confirmed the finding that the probability of choice of OR genes linked to the H element are regulated in cis (*Fuss et al., 2007*), and extended this to over 800 additional OR genes distributed throughout the genome that show little to no evidence in support of *trans*-acting regulation. Instead, the haploid CAST- and B6-derived OR alleles within an F1 are each regulated almost identically as they are in a diploid state within their original genetic backgrounds (*Figure 4F,G*). Our data are inconsistent with *trans*-interactions of multiple enhancers acting additively to regulate the probability of OR choice (*Markenscoff-Papadimitriou et al., 2014*). These *trans* interactions may, however, stabilize or maintain OR singularity after choice has been instructed in cis, and thus could also be necessary for the stereotypic representation of OSNs in a fixed genetic background.

Many existing studies into OR gene choice, especially those utilizing transgenic mouse lines, use animals with a mixed 129/B6 genetic background. The remarkable diversity in the OSN repertoire between these strains (*Figure 2*) suggests caution should be exercised in their interpretation. Here, we created a mouse line that carries the coding sequence of *Olfr2* in the locus of *Olfr1507*, the most frequently selected OR gene, in a pure B6 genetic background. Olfr2-expressing OSNs, which rank 334th across the repertoire in the original B6 strain, are the most abundant OSN subtype in this modified line, demonstrating the critical importance of the genetic context in the regulation of the probability of OR gene choice. Curiously, we also observed that ~10% of other OSN types show comparatively subtle but reproducible differences in abundance. The mechanism underlying these differences is unclear. One plausible hypothesis is that the transposition of *Olfr2* OSNs to a different olfactory zone alters the dynamics and spatial organization of the glomeruli in the olfactory bulb. This in turn may impact on the proliferation and survival of proximal OSN subtypes. Alternatively, stabilizing *trans* interactions could be affected by the OR gene swap, resulting in slightly altered repertoires for some OSN subtypes. In this case, the sequence of an OR receptor gene would influence the relative abundance of other OSN subtypes, but our *Olfr2 > Olfr1507* swap experiment suggests the sequence of an OR receptor does not dictate the abundance of its own OSN subtype.

## Odor-mediated plasticity in the olfactory system

The main olfactory epithelium regenerates throughout the life of an animal. It has been suggested that activity-mediated mechanisms may shape the olfactory system by increasing OSN survival (*François et al., 2013*; *Santoro and Dulac, 2012*; *Watt et al., 2004*; *Zhao and Reed, 2001*), although other studies have found that the number of specific OSN subtypes decrease or are unaffected by odor-exposure (*Cadiou et al., 2014*; *Cavallin et al., 2010*). Each of these studies focused on one or two OSN subtypes and the odor exposure procedures varied significantly in frequency, persistence and length. Here, we took a comprehensive approach, measuring the response of over 1,000 ORs to four odorants, after different types of exposure from 1 week to 6 months. We find that mice living in stable chemical environments maintain the olfactory transcriptomes of their genetically dictated OSN repertoire. However, when frequently recurring odor stimulation is introduced, the abundances of responsive ORs are modified. We propose that this difference is a result of olfactory adaptation in the presence of continuous stimulation. However, other factors may also contribute, for example the continuous exposure odors were likely detected orthonasally, while the intermittently exposure odors were retronasally detected.

The lengthy timeframe for odor-mediated differences to emerge is consistent with modulation of OSN lifespan. It is mechanistically unlikely that odor-activity could influence the probability of OR gene choice, but it could promote OR expression stabilization or singularity. Compared to the dramatic influence of genetic variation on OSN repertoire, odor-evoked changes are subtle, typically less than a two fold change after 6 months of exposure. The limited effect magnitude and long-time scales precluded a more detailed analysis to confirm a correlative alteration in OSN number.

Interestingly, we identified ORs that became more abundant after exposure to specific odorants and, within the same animals, others that became less abundant. Both types were marked by phosphorylation of the S6 ribosomal subunit, a feature of activated OSNs (*Jiang et al., 2015*), indicating that the differential expression is mediated by OSN subtype-specific olfactory stimulation. This may explain why very different conclusions were drawn from previous exposure studies on a small number of ORs (*Cadiou et al., 2014*; *Cavallin et al., 2010*; *François et al., 2013*; *Watt et al., 2004*). Short-term odor exposures (30 min to 24 hr) result in a temporary down-regulation of activated OR mRNA (*von der Weid et al., 2015*), presumably as part of the olfactory adaptation process. It is possible that our analyses are capturing this dynamic short-term response in addition to changes in OSN numbers resulting from long-term exposures. We could not identify any phylogenetic or chromosomal predictor of the ORs that responded with contrasting directional effects, and at this time, the logic underpinning the difference in the direction of expression changes remains unexplained.

### An individually unique olfactory nose

Genetic variation has great impact on individual phenotypic traits. Humans differ in up to a third of their OR alleles by functional variation (*Mainland et al., 2014*), which contributes to an individually unique sense of smell (*Secundo et al., 2015*). Segregating OR alleles have been functionally linked to perceptual differences of their odor ligands, by altering intensity, valence or detection threshold (*Jaeger et al., 2013*; *Keller et al., 2007*; *Mainland et al., 2014*; *McRae et al., 2013*; *Menashe et al., 2007*). However, in most cases, these OR coding genetic variants explain only a small proportion of the observed phenotypic variance (reviewed in *Logan [2014]*), suggesting that other factors contribute to individual differences in perception. Recently, it has been demonstrated that increasing the number of a particular OSN subtype in a mouse nose increases olfactory sensitivity to its ligand (*D'Hulst et al., 2016*). Therefore, the very different OSN repertoires present between strains of mice are likely to result in significant phenotypic variation in olfactory thresholds, and thus contribute to the individualization of olfaction.

Although it remains to be determined whether human OSN repertoires are as variable as the mice reported here, an array-based study of OR expression in 26 humans found unequal expression of ORs within and between individual noses (*Verbeurgt et al., 2014*). Moreover, a recent systematic survey of olfactory perception in humans found high levels of individual variability in reporting the intensity of some odors (for example, benzenethiol and 3-pentanone) but not others (*Keller and Vosshall, 2016*). Further, a non-coding variant within an OR cluster associated with insensitivity to 2-heptanone has been shown to be dominant to the sensitive allele (*McRae et al., 2013*). As OR genes are regulated monoallelically, this implies that a 50% reduction in the sensitive OR allele dosage is, in some cases, sufficient to influence perception. On the other hand, because many odorants activate multiple OSN subtypes (*Malnic et al., 1999*), a differential representation of one subtype may have a limited influence on the overall perception of its odor.

Further investigation into the functional consequence of diverse OSN repertoires will be necessary to determine the full extent to which they individualize the sense of smell.

## Materials and methods

### RNA sequencing

Animal experiments were carried out under the authority of a UK Home Office license (80/2472), after review by the Wellcome Trust Sanger Institute Animal Welfare and Ethical Review Board. All mice were housed in single sex groups within individually ventilated cages, with access to food and water *ad libitum*. All WOM samples were obtained from a single animal, except the pup WOM samples, which were the pool of three or four individuals. Details of the strain, age and sex of each animal sequenced can be found in *Supplementary file 1*. MOEs were dissected and immediately homogenized in lysis RLT buffer (Qiagen, Germantown, Maryland). Total RNA was extracted using the RNeasy mini kit (Qiagen) with on-column DNAse digestion, following the manufacturer's protocol. mRNA was prepared for sequencing using the TruSeq RNA sample preparation kit (Illumina, San Diego, California). All RNA sequencing was paired-end and produced 100-nucleotide-long reads.

## RNAseq data analysis

Sequencing data were aligned with STAR 2.3 (*Dobin et al., 2013*) to the GRCm38 mouse reference genome (B6) or to pseudo-genomes created for the different strains using Seqnature (*Munger et al., 2014*) to impute the high-quality variants reported by the Mouse Genomes Project, release v3 (http://www.sanger.ac.uk/science/data/mouse-genomes-project). The parameters used for mapping were as follows: –outFilterMultimapNmax 1000 –outFilterMismatchNmax 4 –outFilter-MatchNmin 100 –alignIntronMax 50000 –alignMatesGapMax 50500 –outSAMstrandField intronMotif –outFilterType BySJout. The annotation used was from the Ensembl mouse genome database version 72 (http://jun2013.archive.ensembl.org/info/data/ftp/index.html), modified to include all reconstructed gene models for OR genes as reported in (*Ibarra-Soria et al., 2014*).

The numbers of fragments uniquely aligned to each gene were obtained using the HTSeq 0.6.1 package (RRID:SCR_005514), with the script htseq-count, mode *intersection-nonempty* (*Anders et al., 2015*). Raw counts were normalized to account for sequencing depth between samples, using the procedure implemented in the DESeq2 package (*Love et al., 2014*). Data analysis, statistical testing and plotting were carried out in R (http://www.R-project.org).

To compare OR expression levels between datasets, normalization to account for the number of OSNs present in the WOM samples was carried out subsequent to depth normalization. For this, we used a method proposed by *Khan et al. (2013)* that uses marker genes known to be stably expressed in mature OSNs only. This allows estimating the proportion of WOM RNA contributed by the OSNs. Five different marker genes were considered: *Omp*, *Adcy3*, *Ano2*, *Cnga2* and *Gnal* (except for the single-cell data, where *Adcy3* and *Ano2* were excluded because they are not expressed in many cells). To normalize for OSN number, we first computed the correlation coefficient between the expression of each marker gene and the total counts in OR genes; those marker genes with strong correlation values were used for normalization. Then, we calculated the geometric mean of all marker genes for each sample. The average of all geometric means was obtained, and divided by each individual mean; this results in the generation of size factors. Finally, the OR normalized counts were multiplied by the corresponding size factor. Normalized OR expression estimates for the three strains, the *Olfr2 > Olfr1507* and the odor-exposed animals are provided in *Figure 2—source data 1*, *Figure 4—source data 1*, *Figure 5—source data 1* and *Figure 6—source data 1*.

## Differential expression analysis

Differential expression analysis was performed with DESeq2 1.8.1 (*Love et al., 2014*) with standard parameters. To test for differential expression (DE) on the OR repertoire the double normalized counts (accounting for OSN number per sample) were provided directly, and the *normalizationFactors* function was used with size factors of 1 to turn off further normalization. Genes were considered significantly DE if they had an adjusted p-value of 0.05 or less. For the cross-fostering dataset, a likelihood ratio test (*nbinomLRT* function in DESeq2) was used to compare the full model genetics+environment+genetics:environment to a reduced one accounting only for the genetics. Detailed results are provided in *Figure 2—source data 1*, *Figure 4—source data 1*, *Figure 5—source data 1* and *Figure 6—source data 1*.

## In situ hybridization

Probes were designed against nine OR genes chosen among genes covering the receptor expression dynamic range and DE between 129 and B6 strains (*Olfr24, Olfr31, Olfr78, Olfr124, Olfr323, Olfr374, Olfr543, Olfr736* and *Olfr1512*). The following gene-specific oligonucleotides were used to amplify by PCR an amplicon of ~1000 bp from each transcript:

*Olfr24*
TGGCTTACGACCGGTTTGTG (for)
GAAATTAATACGACTCACTATAGGGTTTACACAGCCCAGGATCACAG (rev)
*Olfr31*
TTGCTACCTGCTCGTCTCAC
GAAATTAATACGACTCACTATAGGGCTAGCACTCGGGAGGTTGGAG
*Olfr78*
GAGGAAGCTCACTTTTGGTTTGG
GAAATTAATACGACTCACTATAGGGCAGCTTCAATGTCCTTGTCACAG

*Olfr124*
GGTAATATCTCCATTATCCTAGTTTCCC
GAAATTAATACGACTCACTATAGGGTTGACCCAAAACTCCTTTGTTAGTG
*Olfr323*
TATCCAAGGTCACGGAGTTTCAG
GAAATTAATACGACTCACTATAGGGGAGGGCACTTCCTTTCACTCTG
*Olfr374*
TTGACCTCCTACACACGCATC
GAAATTAATACGACTCACTATAGGGCCAAGACTGGACAAGATTTGGTG
*Olfr543*
ATTCATACAGTGGTGGCCCAG
GAAATTAATACGACTCACTATAGGGCTAAGAATTCAACAAGTCATAGCAGC
*Olfr736*
GGCAATTGTGTATGCAGTGTACTG
GAAATTAATACGACTCACTATAGGGCTGTGAAAAGTTCCCATGTACCTG
*Olfr1512*
TACATCCTGACTCAGCTGGGGAACG
GAAATTAATACGACTCACTATAGGGCACATAGTACACAGTAACAATAGTC

The reverse primer in each case includes the T7 RNA polymerase promoter sequence, and both oligos were designed to amplify a fragment with <75% sequence similarity to other OR genes in the mouse genome. Amplicons were purified from the PCR reactions using the Wizard PCR cleanup kit (Promega, Fitchburg, Wisconsin) and used in riboprobe in vitro transcription with T7 RNA polymerase (ThermoFisher Scientific, Waltham, Massachusetts) and DIG-labeled UTP.

WOM from 10-week-old 129 or B6 mice were collected by dissection, fixed for 48 hr in 4% paraformaldehyde in 1x PBS and demineralized for 10 days in 0.45M EDTA pH 8.0/1x PBS. Samples were then cryoprotected in 0.45M EDTA pH 8.0/1x PBS/20% sucrose for 24 hr, before embedding in OCT medium and sectioning on a Leica CM1850 cryostat to produce slides containing 16 μm coronal MOE sections. Slides were air-dried for 10 min, followed by fixation with 4% paraformaldehyde for 20 min, and treated with 0.1M HCl for 10 min. Tissue acetylation proceeded in 250 mL of 0.1M triethanolamine (pH 8.0) with 1 mL of acetic anhydride for 10 min, with gentle stirring. Two washes in $1\times$ PBS were performed between incubations.

Riboprobe hybridization was done with DIG-labeled probes (1000 ng/mL) at 60°C in hybridization buffer (50% formamide, 10% dextran sulfate, 600 mM NaCl, 200 μg/mL yeast tRNA, 0.25% SDS, 10 mM Tris-HCl pH 8.0, $1\times$ Denhardt's solution, 1 mM EDTA pH 8.0) overnight. Post-hybridization washes included one wash in $2\times$ SSC, one wash in $0.2\times$ SSC and one wash in $0.1\times$ SSC at 55°C, for 30, 20 and 20 min, respectively. Tissue permeabilization was performed in $1\times$ PBS, 0.1% Tween-20 for 10 min, followed by two washes in TN buffer (100 mM Tris-HCl pH 7.5, 150 mM NaCl) for 5 min at room temperature, followed by blocking in TNB buffer (100 mM Tris-HCl pH 7.5, 150 mM NaCl, 0.05% blocking reagent [Perkin Elmer, Waltham, Massachusetts]). Slides were then incubated overnight at 4°C with sheep anti-DIG-AP (Roche, Germany) diluted in TNB (1:800), washed in TNT (100 mM Tris-HCl pH 7.5, 150 mM NaCl, 0.5% Tween 20) six times, 5 min each, transferred to alkaline phosphatase buffer (100 mM Tris-HCl pH 9.8, 100 mM NaCl, 50 mM $MgCl_2$, 0.1% Tween 20) twice for 5 min each.

Signal development was performed in the same buffer containing 5% poly-vinyl alcohol (Mowiol MW 31,000, Sigma-Aldrich, St. Louis, Missouri), 50 μg/mL BCIP and 100 μg/mL NBT, until the purple precipitate was clearly visible with minimum background. Due to the large size of each MOE section, we collected serially scanned images with the 'Scan Large Image' function on NIS Elements software (3.22 version, Nikon Instruments), on a motorized upright Nikon Eclipse 90i microscope equipped with a planar PlanFluor 10x/0.30 DIC L/N1 objective (Nikon, Japan). Background correction was applied on individual images with the NIS elements software and stitched together using Image Composite Editor (Microsoft, Redmond, Washington) with no projection. Linear image adjustments were performed on Photoshop, using the 'Brightness and Contrast' and 'Levels' functions, to equalize the background tone across images. OR-expressing cells were counted by visual inspection. For each gene, three to four animals were analyzed; two to four sections were counted for each animal, and the cell counts were collected independently for each MOE side (hemi-section). The mean number of OR-positive cells per section was calculated (from the two hemi-section counts), followed by

calculation of the mean number of OR-positive cells per animal (from the two to four counted sections). Mean and s.e.m. descriptive statistics were then calculated across the three to four animals analyzed.

### Proportional venn diagrams

Venn diagrams with areas proportional to the number of elements represented were created using the eulerAPE version 3 software (*Micallef and Rodgers, 2014*).

### Identification of copy number variable OR genes in the CAST genome

We mined the Mouse Genomes Project data (*Keane et al., 2011*), release v5 (http://www.sanger.ac. uk/science/data/mouse-genomes-project; RRID:IMSR_JAX:000928). Regions with high numbers of het SNPs indicate multiple alleles being mapped to a single locus in the reference genome. The sequences in the C57BL/6J (RRID:IMSR_JAX:000664) genome (GRCm38) for OR genes with het SNPs in CAST were used to construct a neighbor-joining phylogenetic tree using MEGA6 (*Tamura et al., 2013*). From the tree we selected 33 clades that contained the OR genes with highest number of het SNPs. Then we extracted all of the CAST/EiJ whole-genome Illumina sequencing reads produced by the Mouse Genomes Project that were mapped to these loci (http://www. sanger.ac.uk/science/data/mouse-genomes-project) (*Keane et al., 2011*), realigned these to the members of the respective clade, and then extracted the read pairs poorly aligned to known *Olfr* members (>2% mismatch). We created a de novo assembly of these reads to produce a set of contigs with Geneious R7 (*Kearse et al., 2012*). The contigs were scaffolded and gap-filled using the Illumina reads. From the resulting scaffold sequences, we identified putative new alleles in CAST/EiJ (*Supplementary file 1*). The new allele's sequences are reported in *Figure 2—source data 2*.

### Genetic or environmental effects on OR expression

To dissect the influence of the genetic background from the olfactory environment, C57BL/6N and 129S5 four- to eight-cell stage embryos were transferred into F1 (C57BL/6J×CBA) pseudo-pregnant females. One day after birth, the C57BL/6N and 129S5 litters were cross-fostered to C57BL/6N and 129S5 wild-type mothers, respectively. Then, a single pup from the other strain was transferred to the cross-fostered litter (the *alien*). At weaning, animals from the same sex as the alien animal were kept, always in a 4:1 ratio between strains. If not enough animals of the correct sex were available in the litter, surplus animals from other litters were used. At 10 weeks of age, the WOM was collected form the alien and a randomly selected cage-mate, and RNA was extracted and sequenced as described.

### Generation of the *Olfr2 > Olfr1507* mice

CRISPR-Cas9 technology was used to generate double strand breaks on either side of the *Olfr1507* coding sequence and facilitate homologous recombination. Two guideRNAs -with sequences AAAC TAGATACTTGGCTCATAGG and CATATTCTAGACATTGTCATAGG- were produced with the Ambion T7 MEGAshortscript kit and the Cas9 RNA with the Ambion mMessage mMachine T7 Ultra kit (ThermoFisher Scientific) as specified by the manufacturer's protocols. All RNA were purified with Ambion MegaClear columns, eluting with pre-heated (95°C) elution solution. The eluate was then precipitated with ammonium acetate, and resuspended in ultrapure water (Sigma-Aldrich).

For homologous recombination, we produced a DNA vector containing the coding sequence of *Olfr2* and homology arms for the *Olfr1507* locus of ~1 kb. This was cloned into a modified pUC19 backbone via Gibson Assembly. The sequence-verified plasmid was purified with the NucleoBond Xtra Midi Plus EF Kit (ThermoFisher Scientific) following the manufacturer's protocol. The plasmid was digested to remove the backbone and gel-purified with the QIAquick Gel Extraction Kit (Qiagen) following the kit's protocol. The DNA was precipitated with sodium acetate and resuspended in ultrapure water (Sigma-Aldrich). Finally, the DNA was spin through an Ultrafree-MC centrifugal filter (Merck, Germany).

All components were microinjected into the cytoplasm of 112 C57BL/6N zygotes at the following concentrations: 25 ng/µl for each gRNA, 100 ng/µl of Cas9 RNA and 200 ng/µl of vector DNA. Thirty-eight pups were born and four were positive for the homologous recombination event. One of these was the correct substitution, while the others contained several copies of the DNA vector.

To map the RNAseq data from the *Olfr2 > Olfr1507* homozygous mice, we modified the reference B6 mouse genome (GRCm38) to substitute the *Olfr1507* CDS with that of *Olfr2*. Additionally, the *Olfr2* CDS in the endogenous locus was removed to avoid multimapping. All the counts from both the endogenous *Olfr2* UTRs and the modified *Olfr1507* locus were added together and reported as the *Olfr2* counts; *Olfr1507* was set to zero. The WT controls were mapped to the unmodified reference genome. Data processing and DE analysis were performed as previously described.

## Identification of transcription-factor-binding sites

We used the RegionMiner tool from the Genomatix software suite (https://www.genomatix.de/solutions/genomatix-software-suite.html) to identify overrepresented transcription factor binding sites (TFBSs) in the regions 1 kb upstream of the transcription start site of OR genes as annotated in (*Ibarra-Soria et al., 2014*), for all B6, 129 and CAST sequences. We extracted the match details for the matrix families NOLF and HOMF (Matrix Family Library version 9.3), which correspond to Olf1/Ebf1 and homeodomain TFs, respectively. *Ad hoc* perl scripts were used to parse out the core sequence coordinates of each motif match (defined as the central 10 nucleotides of the reported match), and then to compare the results for each promoter across the strains. We identified those OR genes that had differing number of predicted sites. The number and coordinates of the predicted TFBS are provided in *Figure 4—source data 2*.

## Allelic discrimination of the F1 RNAseq data

The RNAseq data from the WOM of B6 x CAST F1 hybrids were obtained from a pre-publication release by the Wellcome Trust Sanger Institute (ERP004533). Data was processed as described above. Total expression estimates were obtained by mapping the RNAseq data to the B6 or pseudo-CAST genomes, with standard parameters. The expression estimates obtained with each genome were very highly correlated (rho = 0.99, p<$2.2 \times 10^{-16}$). Therefore, the data mapped to the B6 reference was used in downstream analyses.

To obtain allele-specific expression estimates, the RNAseq data was mapped to both the B6 and the pseudo-CAST genomes, without mismatches. Therefore, reads that span SNPs could only map to the genome corresponding to the allele they come from. Subsequent analyses were performed on the OR repertoire only. All reads mapped across each SNP were retrieved with SAMtools (RRID: SCR_002105) (*Li et al., 2009*); *ad hoc* perl scripts were used to exclude reads that splice across the SNP or that were not uniquely mapped. Finally, the number of different reads mapping across all SNPs of each gene was obtained.

To normalize for depth of sequencing, the total expression raw data was combined with the estimates from the parental strains, and normalized all together. The OR data was then further normalized to account for the number of OSNs, as described above. The same size factors were used to normalize the expression estimates from SNP positions.

To deconvolve the total expression into allele-specific expression, a ratio of the expression of each allele was obtained from the counts in SNP positions by dividing the counts in B6 over the total counts in B6 and CAST. Then, the total expression normalized counts were multiplied by the ratio to obtain the B6 expression, and to the inverse of the ratio for the CAST-specific expression. Only those genes with normalized counts in SNP positions above the lowest quartile were used (840 OR genes).

To estimate the amount of technical noise inherent to allelic expression estimation in genes with low counts such as the ORs, we created an in silico F1 dataset. To do this, we used SAMtools (*Li et al., 2009*) to downsample the F0 B6 and CAST samples to 50% of the median depth of the F1 samples; we used the three male B6 samples only. We combined each B6 sample with a CAST sample to create an F1 where the contribution from each allele was 50%. We processed the resulting F1 data as described above. The ratio of the two alleles in the F1 data should match the corresponding ratios in the parental calculations, and we can attribute any deviations from unity to technical noise. Thus, we computed the deviation form the x = y line for each gene in the in silico F1 and used it to correct the fold-change estimate between the allelic expression values in the real F1 data. We used the concordance correlation coefficient as a measurement of the correlation of the fold-change estimates between the parental and allelic expression values (*Lin, 1989*). Unlike a regular correlation

coefficient, it corrects for the agreement on the x = y line, where the data should fall if regulation occurs in cis. Furthermore, it provides the parameter $C_b$ which quantifies the deviation of the best fit to the data from the x = y line; $C_b$ = 1 means no deviation. For both the in silico F1 and the real F1 data, the $C_b$ was 0.99, indicating that the OR repertoire as a whole lies on the diagonal.

## Odor-exposure in vivo

B6 mice were exposed to a mix of heptanal, (R)-carvone, eugenol and acetophenone at 1 mM concentration each, diluted in mineral oil (all odorants from Sigma-Aldrich except acetophenone, from Alfa Aesar). For the acute exposure experiments, the odor mix was added to the water bottles of the animals; mineral oil alone was used for controls. Water bottles were replaced twice a week with freshly prepared ones. The exposure started from at least embryonic day (E)14.5 and the WOM was collected from age-matched exposed and control groups at different time-points after the start of the treatment. For the chronic exposure experiments, the odor mixture or mineral oil only, were applied to a cotton ball with a plastic pasteur pipette; these were put into metal tea strainers that were then introduced into the cage of the animals. The cotton ball was replaced fresh daily. The exposure started from birth and the WOM was collected from age-matched exposed and control groups at different time-points after the start of the treatment.

For the follow-up experiments, animals were acutely exposed only to (R)-carvone, or heptanal, or to the combination of both. The final concentration of each odorant was 1 mM. The odorants were directly added to the water bottles, without dilution in mineral oil. Therefore, the controls were kept with pure water. The water bottles were changed twice a week. The exposure started from at least E16.5 and the WOM was collected at 10 weeks of age.

## qRT-PCR expression estimation

For qRT-PCR experiments, RNA from WOM was extracted as previously described. 1 μg of RNA was reversed-transcribed into cDNA using the High-Capacity RNA-to-cDNA kit (Applied Biosystems, Waltham, Massachusetts) with the manufacturer's protocol. Predesigned TaqMan gene expression assays were used on a 7900HT Fast Real-Time PCR System (ThermoFisher Scientific) following the manufacturer's instructions. Mean cycle threshold (Ct) values were obtained from two technical replicates, each normalized to *Actb* using the ΔCt method. Relative quantity (RQ) values were calculated using the formula RQ = $2^{\Delta Ct}$. Differential expression between groups was tested in R, by a two-tailed t-test, with multiple-testing correction by the Benjamini and Hochberg (FDR) method.

## Luciferase assay

For OR response in vitro, a Dual-Glo Luciferase Assay System (Promega) was employed using the previously described method (*Zhuang and Matsunami, 2008*). Modified HEK293T cells, Hana3A cells, were obtained directly from the Matsunami Laboratory. Cell line identity and negative mycoplasma status was confirmed by PCR after the completion of all experiments. Cells were plated on 96-well PDL plates (ThermoFisher Scientific) for transfection with 5 ng/well of RTP1S-pCI (*Saito et al., 2004*; *Zhuang and Matsunami, 2007*), 5 ng/well of pSV40-RL, 10 ng/well pCRE-luc, 2.5 ng/well of M3-R-pCI (*Li and Matsunami, 2011*), and 5 ng/well of plasmids encompassing the six olfactory receptors of interest. (R)-carvone and heptanal (Sigma-Aldrich) were diluted to a 1 mM solution in CD293 (ThermoFisher Scientific) from 1M stocks in DMSO. 24 hr following transfection, we applied 10-fold serial dilutions of each odorant from 1 mM to 1 nM in triplicate. Luminescence was measured after a 4-hr odor stimulation period using a Synergy two plate reader (BioTek, Winooski, Vermont). Transfection efficiency was controlled for by normalizing all luminescence values by the Renilla luciferase activity. The data were fit to a sigmoidal curve and every OR-odorant pair was compared to a vector-only control using an extra sums-of-squares F test (significantly different from empty vector if $p < 0.05$, the s.d. of the fitted log(EC50) was less than one log unit, and the 95% confidence intervals of the top and bottom parameters did not overlap). Data were analyzed with GraphPad Prism 7.00 and R.

## pS6 immunoprecipitation and RNAseq

Three- to four-week old C57BL/6 mice were placed individually into sealed containers (volume ≈ 2.7L) inside a fume hood and allowed to rest for 1 hr in an odorless environment. For odor stimulus, 10 μl odor solution or 10 μl distilled water (control) was applied to 1cm × 1cm filter paper held in a cassette. The cassette was placed into a new mouse container into which the mouse was also transferred, and the mouse was exposed to the odor solution or control for 1 hr. Experiments were performed in triplicates or quadruplicates, and within each replication the experimental and control mice were littermates of the same sex.

Following odor stimulation, the mouse was sacrificed and the OE was dissected in 25 ml of dissection buffer (1 × HBSS ( with Ca$^{2+}$ and Mg$^{2+}$), 2.5 mM HEPES (pH 7.4 adjusted with KOH), 35 mM glucose, 100 μg/ml cycloheximide, 5 mM sodium fluoride, 1 mM sodium orthovanadate, 1 mM sodium pyrophosphate, 1 mM beta–glycerophosphate) on ice. The dissected OE was transferred to 1.35 ml homogenization buffer (150 mM KCl, 5 mM MgCl$_2$, 10 mM HEPES (pH 7.4 adjusted with KOH), 100 nM Calyculin A, 2 mM DTT, 100 U/ml RNasin (Promega), 100 μg/ml cycloheximide, 5 mM sodium fluoride, 1 mM sodium orthovanadate, 1 mM sodium pyrophosphate, 1 mM beta–glycerophosphate, protease inhibitor (Roche, one tablet/10 ml)) and homogenized three times at 250 rpm and nine times at 750 rpm. The homogenate was transferred to a 1.5 ml lobind tube (Eppendorf, Germany), and centrifuged at 4600 rpm for 10 min at 4°C. The supernatant was then transferred to a new 1.5 ml lobind tube, to which 90 μl 10% NP–40 and 90 μl 300 mM DHPC (Avanti Polar Lipids, Alabaster, Alabama) was added. The mixture was centrifuged at 13,000 rpm for 10 min at 4°C. The supernatant was transferred to a new 1.5 ml lobind tube, and mixed with 20 μl pS6 antibody (Cell Signaling Technology, Danvers, Massachusetts). Antibody binding was allowed by incubating the mixture for 1.5 hr at 4°C with rotation. During antibody binding, Protein A Dynabeads (ThermoFisher Scientific, 100 μl/sample) was washed three times with 900 μl beads wash buffer 1 (150mM KCl, 5 mM MgCl$_2$, 10 mM HEPES (pH 7.4 adjusted with KOH), 0.05% BSA, 1% NP–40). After antibody binding, the mixture was added to the washed beads and gently mixed, followed by incubation for 1 hr at 4°C with rotation. After incubation, the RNA-bound beads were washed 4 times with 700 μl beads wash buffer 2 (RNase free water containing 350 mM KCl, 5 mM MgCl$_2$, 10 mM HEPES (pH 7.4 adjusted with KOH), 1% NP–40, 2 mM DTT, 100 U/ml recombinant RNasin (Promega), 100 μg/ml cycloheximide, 5 mM sodium fluoride, 1 mM sodium orthovanadate, 1 mM sodium pyrophosphate, 1 mM beta–glycerophosphate). During the final wash, beads were placed onto the magnet and moved to room temperature. After removing supernatant, RNA was eluted by mixing the beads with 350 μl RLT (Qiagen). The eluted RNA was purified using RNeasy Micro kit (Qiagen). Chemicals were purchased from Sigma if not specified otherwise.

1.5 μl purified RNA was mixed with 5 μl reaction mix (1× PCR buffer (Roche), 1.5 mM MgCl$_2$, 50 μM dNTPs, 2 ng/μl poly–T primer (TATAGAATTCGCGGCCGCTCGCGA TTTTTTTTTTTTTTTTTTTTTTTT), 0.04 U/μl RNase inhibitor (Qiagen), 0.4 U/μl recombinant RNasin (Promega)). This mixture was heated at 65°C for 1 min and cooled to 4°C. 0.3 μl RT mix (170 U/μl Superscript II (ThermoFisher Scientific), 0.4 U/μl RNase inhibitor (Qiagen), 4 U/μl recombinant RNasin (Promega), 3 μg/μl T4 gene 32 protein (Roche)) was added to each tube and incubated at 37°C for 10 min then at 65°C for 10 min. 1 μl ExoI mix (2 U/μl ExoI (NEB), 1× PCR buffer (Roche), 1.5 mM MgCl$_2$) was added to each tube and incubated at 37°C for 15 min then 80°C for 15 min. 5 μl TdT mix (1.25 U/μl TdT (Roche), 0.1 U/μl RNase H (ThermoFisher Scientific), 1× PCR buffer (Roche), 3 mM dATP, 1.5 mM MgCl$_2$) was added to each tube and incubated at 37°C for 20 min then 65°C for 10 min. 3.5 μl of the product was added to 27.5 μl PCR mix (1× LA Taq reaction buffer (TaKaRa, Japan), 0.25 mM dNTPs, 20 ng/μl poly–T primer, 0.05 U/μl LA Taq (TaKaRa)) and incubated at 95°C for 2 min, 37°C for 5 min, 72°C for 20 min, then 16 cycles of 95°C for 30 s, 67°C for 1 min, 72°C for 3 min with 6 s extension for each cycle, and then 72°C for 10 min. The PCR product was purified by gel purification, and 50 ng of the purified product was used for library preparation with Nextera DNA Sample Prep kits (Illumina). Libraries were sequenced on a HiSeq 2000/2500 (12 libraries pooled per lane) to produce 50 base pair single-end reads. The sequencing data have been deposited in the Gene Expression Omnibus database (https://www.ncbi.nlm.nih.gov/geo/) under accession GSE87695.

Short reads were aligned to the mouse reference genome mm10 using Bowtie (*Langmead et al., 2009*). The reads mapped to annotated genes were then counted using BEDTools (*Quinlan and*

*Hall, 2010*); the gene models for OR genes were replaced by those reported in *Ibarra-Soria et al. (2014)*. A rescuing scheme was used as implemented in *Jiang et al. (2015)* (code available at https://github.com/Yue-Jiang/RNASeqQuant (*Jiang, 2017*) with a copy archived at https://github.com/elifesciences-publications/RNASeqQuant). The read count tables were then analyzed using EdgeR (*Robinson et al., 2010*) to identify differentially expressed ORs.

## Acknowledgements

We thank M Khan and P Mombaerts for providing RNA from the *Olfr7* cluster deletion mouse line that was used in *Figure 1B*; G Gurria and SS Gerety for experimental support and insightful discussions; and the Sanger Institute Research Support Facility, microinjection and sequencing pipeline staff for invaluable technical support.

## Additional information

### Funding

| Funder | Grant reference number | Author |
|---|---|---|
| Wellcome | 098051 | Ximena Ibarra-Soria<br>Jingtao Lilue<br>Mairi Kusma<br>Andrea Kirton<br>Luis R Saraiva<br>Thomas M Keane<br>Darren W Logan |
| Fundação de Amparo à Pesquisa do Estado de São Paulo | 09/00473-0 | Fabio Papes |
| Fundação de Amparo à Pesquisa do Estado de São Paulo | 2015/50371-0 | Fabio Papes |
| European Molecular Biology Organization | Young Investigator Award | Darren W Logan |

The funders had no role in study design, data collection and interpretation, or the decision to submit the work for publication.

### Author contributions

XI-S, Conceptualization, Data curation, Formal analysis, Investigation, Visualization, Methodology, Writing—original draft, Writing—review and editing; TSN, JL, YJ, CT, Formal analysis, Investigation, Writing—review and editing; MAAS, PHMN, KI, NRM, LRS, Investigation, Writing—review and editing; MK, AK, Methodology, Writing—review and editing; TMK, HM, JM, Formal analysis, Supervision, Writing—review and editing; FP, Formal analysis, Supervision, Investigation, Writing—review and editing; DWL, Conceptualization, Formal analysis, Supervision, Funding acquisition, Methodology, Writing—original draft, Project administration, Writing—review and editing

### Author ORCIDs

Ximena Ibarra-Soria, http://orcid.org/0000-0002-9387-3841
Jingtao Lilue, http://orcid.org/0000-0002-1958-0231
Luis R Saraiva, http://orcid.org/0000-0003-4079-0396
Hiroaki Matsunami, http://orcid.org/0000-0002-8850-2608
Joel Mainland, http://orcid.org/0000-0002-5056-4598
Fabio Papes, http://orcid.org/0000-0001-5034-4088
Darren W Logan, http://orcid.org/0000-0003-1545-5510

### Ethics

Animal experimentation: The breeding of mice and experimental procedures were carried out under the authority of a UK Home Office license (80/2472), after review and approval by the Animal Welfare and Ethical Review Body of the Wellcome Trust Sanger Institute.

## Additional files

### Supplementary files

• Supplementary file 1. Sample and gene accessions. (1) Information on the sex, age and strain of the samples sequenced, along with the accession numbers for the raw data; and (2) information on the reconstructed OR alleles for the CAST strain.

### Major datasets

The following datasets were generated:

| Author(s) | Year | Dataset title | Dataset URL | Database, license, and accessibility information |
|---|---|---|---|---|
| Ibarra-Soria X, Logan DW | 2014 | Transcriptome of the MOE in odour exposed mice | http://www.ebi.ac.uk/ena/data/view/PRJEB5984 | Publicly available at the EMBL-EBI European Nucleotide Archive (accession no: PRJEB5984) |
| Ibarra-Soria X, Logan DW | 2014 | Transcriptome of the MOE in different mouse strains | http://www.ebi.ac.uk/ena/data/view/PRJEB5013 | Publicly available at the EMBL-EBI European Nucleotide Archive (accession no: PRJEB5013) |
| Ibarra-Soria X, Logan DW | 2014 | Transcriptome of the olfactory system of newborn mice | http://www.ebi.ac.uk/ena/data/view/PRJEB1607 | Publicly available at the EMBL-EBI European Nucleotide Archive (accession no: PRJEB1607) |
| Ibarra-Soria X, Logan DW | 2016 | RNAseq of an Olfr2>Olfr1507 mouse MOE | http://www.ebi.ac.uk/ena/data/view/PRJEB13659 | Publicly available at the EMBL-EBI European Nucleotide Archive (accession no: PRJEB13659) |
| Ibarra-Soria X, Logan DW | 2014 | Transcriptome of the CAST MOE | http://www.ebi.ac.uk/ena/data/view/PRJEB6462 | Publicly available at the EMBL-EBI European Nucleotide Archive (accession no: PRJEB6462) |

The following previously published dataset was used:

| Author(s) | Year | Dataset title | Dataset URL | Database, license, and accessibility information |
|---|---|---|---|---|
| Saraiva LR, Ibarra-Soria X, Logan DW | 2013 | Single cell transcriptomics in olfactory sensory neurons | http://www.ebi.ac.uk/ena/data/view/PRJEB4461 | Publicly available at the EMBL-EBI European Nucleotide Archive (accession no: PRJEB4461) |

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
