## [Decision Letter]

Thank you for submitting your article "Variation in olfactory neuron repertoires is genetically controlled and environmentally modulated" for consideration by *eLife*. Your article has been reviewed by three peer reviewers, and the evaluation has been overseen by a Reviewing Editor and Patricia Wittkopp as the Senior Editor. The reviewers have opted to remain anonymous.

The reviewers have discussed the reviews with one another and the Reviewing Editor has drafted this decision to help you prepare a revised submission.

Summary:

The authors performed a comprehensive analysis for expression levels of over 100 odorant receptors (ORs) in different mouse strains. They use the large data sets to explore individual- and strain-specific variation in OR expression, the putative role of cis-regulatory elements in choice probability, and the influence of long-term environmental exposure to odors on the OR repertoire. The comprehensiveness is the appealing point of this paper, and the study is thorough and well done. Thus, the major advancement in this paper is to expand data that had been available only from a limited number of ORs in previous studies.

Essential revisions:

1) The authors show that OR gene counts may be used as a proxy for OSN number within a single genetic background and infer that this holds for other strains. To make the case that cell number changes (rather than expression levels per cell) vary between strains, the authors need to validate more OR genes by in situ hybridizations in the different strains. Alternatively, single cell RNAseq could be used to demonstrate the variability in expression for neurons expressing the same OR within and across strains. The conclusions of the manuscript are significantly weakened without a more convincing validation that across-species DE can be used to infer species-specific OSN numbers.

2) A major claim of this paper derives from the observation that each allele of F1 hybrids expresses ORs at levels similar to that expressed by the parental strains. This leads to the claim that "choice" requires regulation in cis but not in trans. There are three reasons why this claim is not supported by the evidence. First, although Figure 4 seems more or less linear, there is substantial diversity (in log space) about the equality line, and no r-squared or equivalent metric is calculated. In addition to remediating the statistical deficits in this key figure, the deviation from unity is important and not commented upon; indeed it is unclear how the authors might interpret this deviation conceptually (i.e., how much deviation would "disprove" their hypothesis that regulation is in cis). Second, the broad (and nearly wholly unexplained) influence of an OR sequence on the expression of 10% of the OR repertoire means that there is no way to disambiguate OR sequence effects from enhancer/promoter effects (particularly since many points on 4F fall off the unity line). Third, even accepting that the data show a clear influence of cis elements on choice frequency (which is likely but not definitively shown given the data), these experiments do not afford any insight regarding the potential role for trans-interactions in choice per se; they only show that choice frequency is determined in cis (and leave open the possibility that trans-interactions are permissive for choice and therefore are critical for the singularity of gene expression), which was the prevailing model even before these data were generated. Indeed the authors repeatedly invoke trans-interactions in the paper as a possible explanation for various aspects of the data, despite strong claims that trans-interactions are not important for "choice". Constraining the interpretation of these data (largely through clarifying the language throughout to make clear that by "choice" what is meant is "choice probability") will more fairly contextualize this result. Note, also, that the definitive experiment here is one that was not performed: swapping the cis-elements between species (rather than two OR cDNAs within a species) to show that polymorphisms in cis-elements determine choice frequency.

3) The relevance of odor exposure experiments in Figure 5 and Figure 6 is unclear, as we really don't know how the odor exposures differed from the perspective of the animal nor do we understand the perceptual consequences of these gene expression changes. Furthermore, the interpretation of these experiments is also incomplete – the receptors seem to be bidirectionally modulated (in equal measure), and only a fraction of these receptors are associated with pS6 signals.

4) In Figure 4, the coding sequence swap of Olfr1507 with that of Olfr2 significantly changed the expression of ~10% of OR genes. This broad change in the expression of the OR repertoire due to swapping single OR gene apparently contradicts to the main claim of this paper, cis-elements determine the OR choice frequency. Is it possible that the odd gene expression effects of the CRISPR experiment reflect off-target indel formation in the OR repertoire? Was any targeted genomic sequencing done to address this concern? Is genomic architecture altered in these animals, consistent with a fundamental rearrangement of trans-interactions (which would be very interesting)?; this could be assessed by chromosome conformation capture experiments (4C against Olfr1507 promoter or enhancer or Hi-C), but performing these experiments would be only important if the authors decided to make this the focus of a revised manuscript.

[Editors' note: further revisions were requested prior to acceptance, as described below.]

Thank you for resubmitting your work entitled "Variation in olfactory neuron repertoires is genetically controlled and environmentally modulated" for further consideration at *eLife*. Your revised article has been favorably evaluated by Patricia Wittkopp (Senior editor), a Reviewing editor and one reviewer.

The manuscript has been greatly improved but there are minor textual changes requested to discuss the potential role of coding and non-coding sequences (see below). These text revisions should be straightforward to complete and I do not anticipate that they will cause a significant time delay.

Reviewer #2:

The paper entitled "Variation in olfactory neuron repertoires is genetically controlled and environmentally modulated" is a resubmission from Logan et al. that uses RNASeq and genetics to probe the diversity of OR expression in the olfactory epithelium. This manuscript is substantially improved from the previous submission, and in particular now appropriately characterizes what is learned from the paper. The improved statistical analyses are also very helpful. However, there are a couple small points that are worth revisiting through text revisions before publication of this very nice paper.

Major point:

The authors seem at pains to argue that the primary sequence of each OR does not contribute to the pattern of expression of that OR, but I'm still having trouble seeing the argument clearly. Just to be transparent about my point of view here, I think the authors have done a very nice job of showing that sequences that are linked to the OR are instructive for choice frequency. I just don't see any evidence that rules out OR primary sequences in this process, and indeed see some evidence that it may play a role, for the following reasons:

a) the fact that similar protein sequences are differentially regulated across strains (the main argument leveled here) doesn't rule out the converse – that distinct protein sequences might contribute to differential regulation.b) The attempt to call the differences in OR expression in the receptor swap experiment "subtle" and therefore not important ("the extensive variance…is independent of the coding sequence […] of the OR […]") is really a qualitative judgment rather than a quantitative argument, especially given that in the same manuscript similar effect sizes are argued to be relevant in the context of odor exposures, and given the argument in the discussion that fold changes in OR expression have perceptual meaning.c) Most importantly, unless I really am missing something, the F1 analysis doesn't distinguish coding from non-coding effects, as these are linked in the intercross.

I think some additional circumspection is merited here. The authors have the benefit that the model they favor – that cis elements are instructive and that protein sequences don't matter that much if at all – is likely to be right. It is just that given the data assembled here this argument seems less definitive than the authors seek to make it.

---

## [Author Response]

*Essential revisions:*

*1) The authors show that OR gene counts may be used as a proxy for OSN number within a single genetic background and infer that this holds for other strains. To make the case that cell number changes (rather than expression levels per cell) vary between strains, the authors need to validate more OR genes by in situ hybridizations in the different strains.*

Although it might appear there are a multitude of ORs to select from, identifying appropriate receptors to compare by *in situ* hybridization is surprisingly difficult from a technical perspective, because it is imperative that:

1) They must be selected from across the dynamic range of expression (yet the vast majority of receptors are expressed at very low levels, see Figure 1)

2) The probes must not cross-hybridize to any other receptors within each strain (yet most ORs have high sequence identity to at least one other OR)

3) The probes must equally hybridize to the same receptor transcript between strains (yet there are over 15,000 distinct sequence variations in ORs between the strains we tested)

4) The ORs must be differentially expressed between strains by RNAseq (yet two thirds of the repertoire are not, see Figure 2).

Through a process of elimination we identified 6 receptor genes that best met these criteria across two strains (B6 and 129). When we analysed these data we found almost perfect correlations between OSN counts and RNAseq expression data within a strain for both B6 (Figure 1) and 129. To make more evident that this holds across strains, we have explicitly stated this in the text and now present in Figure 2—figure supplement 2B the same plot shown in Figure 1 but with the data from 129 animals. We further showed very strong correlations between the fold change in the expression and OSN counts across strains (Figure 2). We believe these data strongly support the conclusion that OSN counts differ between strains and that RNAseq approximates these differences.

To further supplement this analysis we have now identified and carried out *in situ* hybridization on three further receptor genes, including *Olfr78* which is not significantly different in the RNAseq data and *Olfr124* and *Olfr1512* that have smaller differences between the strains. All data points show a strong linear relationship and the fold-change values estimated with either technique are strongly correlated (Spearman’s rho = 0.98, p = 0.00005) These additional data confirm our correlations, and now show, in Figure 2, that for 9 different ORs the differences in RNAseq expression between strains are a proxy for the differences in OSN numbers.

*Alternatively, single cell RNAseq could be used to demonstrate the variability in expression for neurons expressing the same OR within and across strains. The conclusions of the manuscript are significantly weakened without a more convincing validation that across-species DE can be used to infer species-specific OSN numbers.*

We agree that single cell RNAseq would be an elegant and powerful tool to further address this question. It is something we have considered, however our previous experience in single cell RNAseq of OSNs and VSNs (Saraiva, Ibarra-Soria et al. 2015, Untiet et al. 2016) taught us that it would be a prohibitively lengthy and expensive undertaking. All the existing genetic tools that fluorescently label specific OSN types were generated on mixed genetic backgrounds, making them unsuitable for these experiments. Thus taking a targeted approach would require the de novo generation of many new mouse lines. An approach involving random selection of neurons would require many thousands of OSNs to be hand-picked and sequenced to ensure the same OSN types were represented enough times from each strain. We concluded that this approach is not feasible with the technology currently available.

2) A major claim of this paper derives from the observation that each allele of F1 hybrids expresses ORs at levels similar to that expressed by the parental strains. This leads to the claim that "choice" requires regulation in cis but not in trans. There are three reasons why this claim is not supported by the evidence. First, although Figure 4 seems more or less linear, there is substantial diversity (in log space) about the equality line, and no r-squared or equivalent metric is calculated.

We plotted the axes in Figure 4 in such a way that if OR choice is only regulated in *cis*, the data would be distributed diagonally along a 45 degree line. If choice is only regulated in *trans*, the data would be distributed vertically along a 90 degree line. As the reviewers note, collectively the data is distributed diagonally along the equality (45 degree) line. We have now indicated this as a metric in Figure 4, as requested, by calculating the concordance correlation coefficient (ccc=0.86) which, in contrast to r-squared, adjusts for agreement on the x=y line (a value of 1 equals perfect concordance (Lin, 1989)). These data supports our conclusion that, taken as a whole, the OR repertoire is regulated in *cis*. The novelty of this analysis is that it captures over 800 ORs, whereas previous studies have focused on single or a handful of ORs. This scale is important because, as the reviewers also note, there is individual OR diversity from the equality line; thus drawing conclusions from any single or small number of ORs could be misleading.

*In addition to remediating the statistical deficits in this key figure, the deviation from unity is important and not commented upon; indeed it is unclear how the authors might interpret this deviation conceptually (i.e., how much deviation would "disprove" their hypothesis that regulation is in cis).*

To ascertain whether the observed deviation from the equality line is biologically meaningful or technical noise from the complexity of distinguishing alleles from each other, we carried out an in silico pseudo F1-cross. In essence we combined RNAseq read data from B6 and CAST F0 mice then down-sampled by 50% to approximate expression from one allele. We then carried out the same analysis as in Figure 4. We now show in Figure 4—figure supplement 1 that in this in silico F1 the data falls largely along the 45 degree line but, as with the real F1 data, there is still some deviation (ccc=0.95). This variance can be attributed to technical noise, since we know the data contains a 50:50 proportion of the parental data. We computed the deviation from the equality line for each gene and used this measurement to correct the real F1 data. The resulting plot is now shown in Figure 4—figure supplement 1. We have used the concordance correlation coefficient (ccc) to measure the correlation of the fold-change estimates, while accounting for agreement on the x=y line; this is further quantified by the Cb parameter, which indicates how far the best fit to the data deviates from the 45 degree line. The corrected real data has a ccc = 0.92 and a Cb = 0.999 which indicates that the data, as a whole, is best fit by the x=y line and thus shows a pattern entirely consistent with regulation in cis. Our interpretation of this additional analysis is that the observed deviation is technical noise but we cannot exclude a very small contribution from trans-acting elements. We have now indicated this in subsection “OSN Diversity Profiles are Independent of OR Function and are Controlled in cis”.

*Second, the broad (and nearly wholly unexplained) influence of an OR sequence on the expression of 10% of the OR repertoire means that there is no way to disambiguate OR sequence effects from enhancer/promoter effects (particularly since many points on 4F fall off the unity line).*

We acknowledge that the influence of a single OR sequence on differential representation of 10% of the OR repertoire is consistent with trans regulation (subsection “OSN Diversity Profiles are Independent of OR Function and are Controlled in cis”). However, the size of the enhancer/promoter effect (a 47 fold difference) compared to the OR sequence effects (<2 fold) does disambiguate them, suggesting that any *trans* influence is subtle (Figure 4). Our hypothesis to explain this unexpected observation, which we have now further elaborated on in subsection “The MOE is a genetically-determined mosaic of OSN subtypes”, is that the artificial transposition of an OSN type to a different olfactory zone (the consequence of the OR swap) will undoubtedly influence the dynamics and spatial orientation of its projections to form glomeruli in the olfactory bulb. The impact of this transposition on projection routes, and the size, shape and location of neighbouring glomeruli could conceivably influence the proliferation and survival of other OSNs during development, and thus explain the subtle differences in OR expression we observe. This process would occur post-“choice” and thus be independent of *cis/trans* regulation of OR expression. Nevertheless, we accept that extensive, albeit subtle, *trans* effects is an alternative hypothesis. This could also potentially explain the deviations from the line of equality in Figure 4. We have altered our conclusions to address this possibility.

*Third, even accepting that the data show a clear influence of cis elements on choice frequency (which is likely but not definitively shown given the data), these experiments do not afford any insight regarding the potential role for trans-interactions in choice per se; they only show that choice frequency is determined in cis (and leave open the possibility that trans-interactions are permissive for choice and therefore are critical for the singularity of gene expression), which was the prevailing model even before these data were generated. Indeed the authors repeatedly invoke trans-interactions in the paper as a possible explanation for various aspects of the data, despite strong claims that trans-interactions are not important for "choice". Constraining the interpretation of these data (largely through clarifying the language throughout to make clear that by "choice" what is meant is "choice probability") will more fairly contextualize this result.*

We disagree that our data does not show a definitive influence of *cis* elements on choice frequency. We are unaware of another credible hypothesis to explain the collective distribution of the data in Figure 4, other than through *cis* regulation of OR choice. Indeed, the same approach has been used to distinguish *cis* from *trans* regulation in a number of other studies (e.g. Goncalves et al. 2012; Emmerson et al. 2010). We acknowledge that trans-interactions may indeed stabilize or maintain OR singularity, and are also clear where our data is consistent with, and inconsistent with, previously proposed models (subsection “The MOE is a genetically-determined mosaic of OSN subtypes”). As indicated above, we have now added additional text to temper our conclusions and discuss the possibility of trans-regulation explaining the observations from Figure 4. We have also clarified our language around the use of the word “choice” throughout the manuscript, as suggested. Together we believe these edits more fairly contextualise our results.

*Note, also, that the definitive experiment here is one that was not performed: swapping the cis-elements between species (rather than two OR cDNAs within a species) to show that polymorphisms in cis-elements determine choice frequency.*

We agree that, conceptually, this would be an ideal experiment. However unlike the cDNA sequence of an OR, which has a clear functional beginning and end point, the promoters and enhancers that control OR choice are ill-defined. We therefore felt we could not clearly interpret the results of a promoter-swap experiment as the possibility would always remain that only part of the promoter was transposed. We investigated swapping one of the few known, better defined OR enhancer elements between strains. However bioinformatic analysis revealed that, between the strains, there was substantial differences in the distance between the enhancer elements and their neighbouring OR cluster. Considering strains we have expression data for, there appears to be an association between distance of an element from the cluster and the abundances of the OR expression within that cluster. Given these data, we felt it is likely that both *cis*-element sequence and its distance-from-OR were potential factors in controlling *cis* regulation. We did not consider it technically feasible to genetically engineer the multiple mouse lines necessary to control for these multiple variables.

3) The relevance of odor exposure experiments in Figure 5 and Figure 6 is unclear, as we really don't know how the odor exposures differed from the perspective of the animal.

The purpose of the odor exposure experiments to our study were two-fold. Firstly, we were aware that different strains of mice have different odortypes, and that this could influence OSN subtype repertoire. Hence we first carried out the experiments in Figure 3 to test this directly. We clearly showed that long term exposure to different strain odor-types did not influence the OSN repertoire (Figure 3). This result appeared to directly conflict the findings by Santoro and Dulac (*eLife*, 2012), who reported a mechanism through which long term odor exposure should theoretically alter OSN subtype representation in an OR-specific manner. To reconcile these apparent differences we carried out the experiments in Figure 5, to establish that the exposure paradigm critically determines the effect on OR abundance, and Figure 6, to demonstrate this effect is ligand and OR-specific.

Secondly, other previous studies had reported that long-term odor exposure had variously increased, decreased or had no effect on the number of OSN subtypes. However each of these studies used different exposure methods and measured only a single, or a handful of ORs. Our approach permits, for the first time, analysis of the entire repertoire in parallel. As the reviewers note, we demonstrate reproducible bidirectional changes in an OR-specific manner (Figure 5, Figure 6). We also show that changing the exposure paradigm alters whether these changes occur or are maintained (Figure 5). Thus we have reconciled the conflicting literature by exploiting the power afforded by large-scale analysis. We agree that we do not know how the odor exposures differ in terms of perception, and acknowledge this in the text. However, we do not consider this to be critical for the purposes of the experiment.

*Nor do we understand the perceptual consequences of these gene expression changes.*

In this manuscript, we do not functionally investigate the perceptual consequences of either the genetically-instructed variance in OSN repertoire nor the influence of environmental exposure. Because individual odors activate multiple ORs, it is near impossible to devise controlled experiments to investigate how differences in individual OSN subtype representation alter odor perception using wild-type strains. We anticipated that the *Olfr2>Olfr1507* line could be uniquely suited for this purpose, but the unexpected changes in the representation of over 100 other OSN subtypes meant its usefulness for phenotypic studies was compromised. However, recent work from D’Hulst et al. 2016 has demonstrated that odor sensitivity increases when the number of responsive OSNs in the nose is greater. Thus we speculate on how the differences we catalogue in this study may alter the perception of smell (subsection “An individually unique olfactory nose”).

Furthermore, the interpretation of these experiments is also incomplete – the receptors seem to be bidirectionally modulated (in equal measure).

We acknowledge that the bidirectional modulation is mechanistically puzzling (although not inconsistent with the existing literature). Despite our best efforts we have been unable to find any pattern or logic that may distinguish the receptors that are up-regulated from those that are down-regulated, thus we do not have a credible hypothesis to test. Moreover, the time-frames involved in each long-term exposure experiment prohibit using this technique as a fishing expedition. For example, the completion of the experiments in Figure 5 and Figure 6 took a number of years.

*And only a fraction of these receptors are associated with pS6 signals.*

Indeed, 55% of them were associated with pS6 signals, a highly statistically significant enrichment (p=0.0001). A number of other ORs identified in each of the experiments were just above the p-value threshold in the other (examples of this can be seen just below the dotted line in Figure 6), thus marginally relaxing the somewhat arbitrary p-value cut-offs would increase the proportion of associated ORs. Moreover we do not consider the lack of a complete association between receptors identified using the two techniques to be unexpected, because the odor exposure and RNA processing methods used for each experiment differed. We purposely applied the exposure method previously reported for pS6 immunoprecipitation (Jian et al. 2015) as it had been optimised for receptor deorphanisation, and the goal of this experiment was to prove that the DE ORs in the long term experiments were linked to direct odor activation. The highly significant statistical association between the ORs identified in the two methods supports this conclusion.

4) In Figure 4, the coding sequence swap of Olfr1507 with that of Olfr2 significantly changed the expression of ~10% of OR genes. This broad change in the expression of the OR repertoire due to swapping single OR gene apparently contradicts to the main claim of this paper, cis-elements determine the OR choice frequency.

As detailed in our response to point 2.3 above, we believe there are other possible explanations for this observation, especially when considered in the context of the other data in our manuscript. Nevertheless, we agree that a relatively subtle effect on *trans* regulation is an alternative hypothesis and have altered our conclusions to address this possibility.

*Is it possible that the odd gene expression effects of the CRISPR experiment reflect off-target indel formation in the OR repertoire? Was any targeted genomic sequencing done to address this concern?*

Although we did not carry out additional targeted sequencing, our analysis suggests this possibility is unlikely. When designing our CRISPR guide RNAs (gRNAs) we identified the coordinates of potential off-target binding sites; the gRNAs we chose showed very few off-targets in exonic regions and none in other OR genes. We have now further applied two more tools to predict potential off-targets –CCTop (Stemmer et al. PLoS One 2015) and Breaking-Cas (Olvieros et al. NAR 2016) – and obtained 84 and 193 for the upstream gRNA (u-gRNA) and 132 and 206 for the downstream gRNA (d-gRNA), respectively:

CCTopBreaking-Casu-gRNAd-gRNAu-gRNAd-gRNAtotal84132193206in genes38429183in exons2121in OR exons0000within 10kb of OR genes0112

Only three OR genes have a predicted off-target site within 10kb and of these, only one (*Olfr1258*) is significantly differentially expressed. In this case, the predicted off-target cleavage site lies within the 3’ UTR; we have inspected the sequencing reads spanning the predicted site in all the Olfr2>Olfr1057 animals and we did not identify any evidence of edits. In addition, we inspected the sequence reads spanning the three other off-target sites within exons and again found no edits were present. Thus, we consider very unlikely that off-target effects could explain the observed expression changes in OR genes.

*Is genomic architecture altered in these animals, consistent with a fundamental rearrangement of trans-interactions (which would be very interesting)?; this could be assessed by chromosome conformation capture experiments (4C against Olfr1507 promoter or enhancer or Hi-C), but performing these experiments would be only important if the authors decided to make this the focus of a revised manuscript.*

We cannot rule out alterations to the genomic architecture have occurred upon the swap of the coding sequences of these OR genes. However, the proposed experiments are technically challenging because it would be necessary to isolate the OSN population from the other cell types in the WOM. The *Olfr2>Olfr1507* line does not carry an OSN-subtype specific or a pan-OSN marker to facilitate this, and we are not aware of an appropriate marker line in a pure B6 genetic background that we could inter-cross with.

[Editors' note: further revisions were requested prior to acceptance, as described below.]

*Reviewer #2:*

*The paper entitled "Variation in olfactory neuron repertoires is genetically controlled and environmentally modulated" is a resumbission from Logan et al. that uses RNASeq and genetics to probe the diversity of OR expression in the olfactory epithelium. This manuscript is substantially improved from the previous submission, and in particular now appropriately characterizes what is learned from the paper. The improved statistical analyses are also very helpful. However, there are a couple small points that are worth revisiting through text revisions before publication of this very nice paper.*

*Major point:*

*The authors seem at pains to argue that the primary sequence of each OR does not contribute to the pattern of expression of that OR, but I'm still having trouble seeing the argument clearly. Just to be transparent about my point of view here, I think the authors have done a very nice job of showing that sequences that are linked to the OR are instructive for choice frequency. I just don't see any evidence that rules out OR primary sequences in this process, and indeed see some evidence that it may play a role, for the following reasons:*

*a) the fact that similar protein sequences are differentially regulated across strains (the main argument leveled here) doesn't rule out the converse – that distinct protein sequences might contribute to differential regulation.b) The attempt to call the differences in OR expression in the receptor swap experiment "subtle" and therefore not important ("the extensive variance […] is independent of the coding sequence […] of the OR […]") is really a qualitative judgment rather than a quantitative argument, especially given that in the same manuscript similar effect sizes are argued to be relevant in the context of odor exposures, and given the argument in the discussion that fold changes in OR expression have perceptual meaning.*

We recognise the point the reviewer makes from the way we phrased our interpretation. We believe our data, particularly the analysis of the *Olfr2>Olfr1507* animals in Figure 4, strongly argues that the primary sequence/function of the OR expressed in a particular subtype does not instruct the abundance of that particular subtype. In other words, when *Olfr2* is expressed in (what would otherwise be) *Olfr1507*-expressing neurons, their abundance does not change to the levels of *Olfr2*-expressing neurons. The fact that identical (not similar) protein sequences are differentially regulated across strains is consistent with this conclusion.

However, we agree that the differences in other ORs expressed in the *Olfr2>Olfr1507* animals is consistent with the *Olfr2* primary sequence influencing the choice of other ORs, but importantly not *Olfr2* itself. We realise that we did not make clear this distinction in the text, and have now done so at multiple points.:

In subsection “OSN Diversity Profiles are Independent of OR Function and are Controlled in *cis*.”: “Together these results suggest that the proportion of each OSN subtype is not dependent on its endogenous OR receptor activity.”

In the same subsection: “the extensive variance in OSN subtype composition we observe within and between in mice is determined by the wider genetic architecture of the animal, and is independent of the coding sequence and function of the OR protein each subtype expresses.

Discussion, first paragraph: “OSN diversity is predominantly controlled by genetic elements that act in *cis*, and the abundance of a given OSN subtype is not affected by receptor the sequence or function of the receptor it expresses, in a sustained olfactory environment.

Subsection “The MOE is a genetically-determined mosaic of OSN subtypes.”: “In this case the sequence of an OR receptor gene would influence the relative abundance of other OSN subtypes, but our *Olfr2>Olfr1507* swap experiment suggests the sequence of an OR receptor does not dictate the abundance of its own OSN subtype.”

We used the term “subtle” only to compare it to the scale of the effect observed in the swapped receptor, not to diminish the potential functional consequences. Indeed, we also use “subtle” to describe the environmental effect in our Abstract. To make this clearer, we now use “comparatively subtle” instead.

*c) Most importantly, unless I really am missing something, the F1 analysis doesn't distinguish coding from non-coding effects, as these are linked in the intercross.*

Reviewer #2 is correct, the F1 analysis does not provide insight into coding from non-coding effects.